# Tree! I am no Tree! I am a Low Dimensional Hyperbolic Embedding

**Rishi Sonthalia**\*
Department of Mathematics
University of Michigan
Ann Arbor, MI, 48104
rsonthal@umich.edu

**Anna C. Gilbert**
Department of Statistics and Data Science
Yale University
New Haven, CT, 06510
anna.gilbert@yale.edu

## Abstract

Given data, finding a faithful low-dimensional hyperbolic embedding of the data is a key method by which we can extract hierarchical information or learn representative geometric features of the data. In this paper, we explore a new method for learning hyperbolic representations by taking a metric-first approach. Rather than determining the low-dimensional hyperbolic embedding directly, we learn a tree structure on the data. This tree structure can then be used directly to extract hierarchical information, embedded into a hyperbolic manifold using Sarkar's construction [38], or used as a tree approximation of the original metric. To this end, we present a novel fast algorithm TREEREP such that, given a $\delta$-hyperbolic metric (for any $\delta \geq 0$), the algorithm learns a tree structure that approximates the original metric. In the case when $\delta = 0$, we show analytically that TREEREP exactly recovers the original tree structure. We show empirically that TREEREP is not only many orders of magnitude faster than previously known algorithms, but also produces metrics with lower average distortion and higher mean average precision than most previous algorithms for learning hyperbolic embeddings, extracting hierarchical information, and approximating metrics via tree metrics.

## 1 Introduction

Extracting hierarchical information from data is a key step in understanding and analyzing the structure of the data in a wide range of areas from the analysis of single cell genomic data [26], to linguistics [13], computer vision [25] and social network analysis [41]. In single cell genomics, for example, researchers want to understand the developmental trajectory of cellular differentiation. To do so, they seek techniques to visualize, to cluster, and to infer temporal properties of the developmental trajectory of individual cells.

One way to capture the hierarchical structure is to represent the data as a tree. Even simple trees, however, cannot be faithfully represented in low dimensional Euclidean space [29]. As a result, a variety of remarkably effective hyperbolic representation learning methods, including Nickel and Kiela [30, 31], Sala et al. [36], have been developed. These methods learn an embedding of the data points in hyperbolic space by first solving a non-convex optimization problem and then extracting the hyperbolic metric that corresponds to the distances between the embedded points. These methods are successful because of the inherent connections between hyperbolic spaces and trees. They do not, however, come with rigorous geometric guarantees about the quality of the solution. Also, they are slow.

---

In this paper, we present a metric first approach to extracting hierarchical information and learning hyperbolic representations. The important connection between hyperbolic spaces and trees suggests that the correct approach to learning hyperbolic representations is the metric first approach. That is, first, learn a tree that essentially preserves the distances amongst the data points and then embed this tree into hyperbolic space.[2] More generally, the metric first approach to metric representation learning is to build or to learn an appropriate metric first by constructing a discrete, combinatorial object that corresponds to the distances and then extracting its low dimensional representation rather than the other way around.

The quality of a hyperbolic representation is judged by the quality of the metric obtained. That is, we say that we have a good quality representation if the hyperbolic metric extracted from the hyperbolic representation is, in some way, faithful to the original metric on the data points. We note that finding a tree metric that approximates a metric is an important problem in its own right. Frequently, we would like to solve metric problems such as transportation, communication, and clustering on data sets. However, solving these problems with general metrics can be computationally challenging and we would like to approximate these metrics by simpler, tree metrics. This approach of approximating metrics via simpler metrics has been extensively studied before. Examples include dimensionality reduction [24] and approximating metrics by simple graph metrics [3, 32].

To this end, in this paper, we demonstrate that methods that learn a tree structure first outperform methods that learn hyperbolic embeddings directly. Additionally, we have developed a novel, extremely fast algorithm TREEREP that takes as input a $\delta$-hyperbolic metric and learns a tree structure that approximates the original metric. TREEREP is a new method that makes use of geometric insights obtained from the input metric to infer the structure of the tree. To demonstrate the effectiveness of our method, we compare TREEREP against previous methods such as Abraham et al. [1] and Saitou and Nei [35] that also recover tree structures given a metric. There is also significant literature on approximating graphs via (spanning) trees with low stretch or distortion, where the algorithms take as input graphs, *not metrics*, and output trees that are subgraphs of the original. We also compare against such algorithms [2, 11, 16, 33]. We show that when we are given only a metric and not a graph, then even if we use a nearest neighbor graph or treat the metric as a complete graph TREEREP is not only faster, but produces better results than [1, 2, 3, 11, 33] and comparable results to [35].

For learning hyperbolic representations, we demonstrate that TREEREP is over 10,000 times faster than the optimization methods from Nickel and Kiela [30, 31], and Sala et al. [36] while producing better quality results in most cases. This extreme decrease in time, with no loss in quality, is exciting as it allows us to extract hierarchical information from much larger data sets in single-cell sequencing, linguistics, and social network analysis - data sets for which such analysis was previously unfeasible.

The rest of the paper is organized as follows. Section 2 contains the relevant background information. Section 3 presents the geometric insights and the TREEREP algorithm. In Section 4, we compare TREEREP against the methods from Abraham et al. [1], Alon et al. [2], Chepoi et al. [11], Prim [33] and Saitou and Nei [35] in approximating metrics via tree metrics and against methods from Nickel and Kiela [30, 31] and Sala et al. [36] for learning low dimensional hyperbolic embedding. We show that the methods that learn a good tree to approximate the metric, in general, find better hyperbolic representations than those that embed into the hyperbolic manifold directly.

## 2 Preliminaries

The formal problem that our algorithm will solve is as follows[3].

**Problem 1.** *Given a metric $d$ find a tree structure $T$ such that the shortest path metric on $T$ approximates $d$.*

**Definition 1.** *Given a weighted graph $G = (V, E, W)$ the shortest path metric $d_G$ on $V$ is defined as follows: $\forall u, v \in V$, $d_G(u, v)$ is the length of the shortest path from $u$ to $v$.*

$\delta$**-Hyperbolic Metrics.** Gromov introduced the notion of $\delta$-hyperbolic metrics as a generalization of the type of metric obtained from negatively curved manifolds [20].

**Definition 2.** *Given a space $(X, d)$, the Gromov product of $x, y \in X$ with respect to a base point $w \in X$ is*

$$(x, y)_w := \frac{1}{2} \left( d(w, x) + d(w, y) - d(x, y) \right).$$

*The Gromov product is a measure of how close $w$ is to the geodesic $g(x, y)$ connecting $x$ and $y$.*

**Definition 3.** *A metric $d$ on a space $X$ is a $\delta$-hyperbolic metric on $X$ (for $\delta \geq 0$), if for every $w, x, y, z \in X$ we have that*

$$(x, y)_w \geq \min \left( (x, z)_w, (y, z)_w \right) - \delta. \tag{2.1}$$

*In most cases we care about the smallest $\delta$ for which $d$ is $\delta$-hyperbolic.*

An example of a $\delta$-hyperbolic space is the hyperbolic manifold $\mathbb{H}^k$ with $\delta = \tanh^{-1} \left( 1/\sqrt{2} \right)$ [9].

**Definition 4.** *The hyperboloid model $\mathbb{H}^k$ of the hyperbolic manifold is $\mathbb{H}^k = \{ x \in \mathbb{R}^{k+1} : x_0 > 0, x_0^2 - \sum_{i=1}^{k} x_i^2 = 1 \}$.*

An important case of hyperbolic metrics is when $\delta = 0$. One important property of such metrics is that they come tree spaces.

**Definition 5.** *A metric $d$ is a tree metric if there exists a weighted tree $T$ such that the shortest path metric $d_T$ on $T$ is equal to $d$.[4]*

**Definition 6.** *Given a discrete graph $G = (V, E, W)$ the metric graph $(X, d)$ is the space obtained by letting $X = E \times [0, 1]$ such that for any $(e, t_1), (e, t_2) \in X$ we have that $d((e, t_1), (e, t_2)) = W(e) \cdot |t_1 - t_2|$. This space is called a tree space if $G$ is a tree. Here $E \times \{0, 1\}$ are the nodes of $G$.*

**Definition 7.** *Given a metric space, $(X, d)$, two points $x, y \in X$, and a continuous function $f : [0, 1] \to X$, such that $f(0) = x$, $f(1) = y$, and there is a $\lambda$ such that $d(f(t_1), f(t_2)) = \lambda |t_1 - t_2|$, the geodesic $g(x, y)$ connecting $x$ and $y$ is the set $f([0, 1])$.*

**Definition 8.** *A metric space $T$ is a tree space (or a $\mathbb{R}$-tree) if any pair of its points can be connected with a unique geodesic segment, and if the union of any two geodesic segments $g(x, y), g(y, z) \subset T$ having the only endpoint $y$ in common, is the geodesic segment $g(x, z) \subset T$.*

There are multiple definitions of a tree space. However, they are all connected via their metrics. Bermudo et al. [4] tells us that a metric space is 0-hyperbolic if and only if it is an $\mathbb{R}$-tree or a tree space. This result lets us immediately conclude that Definitions 6 and 8 are equivalent. Similarly, Definition 1 implies that Definition 5 and 6 are equivalent. Hence all three definitions of tree spaces are equivalent. We note that trees are 0-hyperbolic, and that $\delta = \infty$ corresponds to an arbitrary metric. Thus, $\delta$ is a heuristic measure for how close a metric is to a tree metric.

**Trees as Hyperbolic Representation.** The problem that is looked at by [36, 30, 31] is the problem of learning hyperbolic embeddings. That is, given a metric $d$, learn an embedding $X$ in some hyperbolic space $\mathbb{H}^k$. We, however, are proposing that if we want to learn a hyperbolic embedding, then we should instead learn a tree. In many cases, we can think of this tree as the hyperbolic representation. However, if we do want coordinates, this can be done as well.

Sala et al. [36] give an algorithm that is a modification of the algorithm in Sarkar [38] that can, in linear time, embed any weighted tree into $\mathbb{H}^k$ with arbitrarily low distortion (if scaling of the input metric is allowed). The analysis in Sala et al. [36] quantifies the trade-offs amongst the dimension $k$, the desired distortion, the scaling factor and the number of bits required to represent the distances in $\mathbb{H}^k$. We use these results to consider trees as hyperbolic representations. One possible drawback of this approach is that we may need a large number of bits of precision. Recent work, however, such as Yu and De Sa [43] provides a solution to this issue.

## 3 Tree Representation

To solve Problem 1 we present an algorithm TREEREP such that Theorem 1 holds.

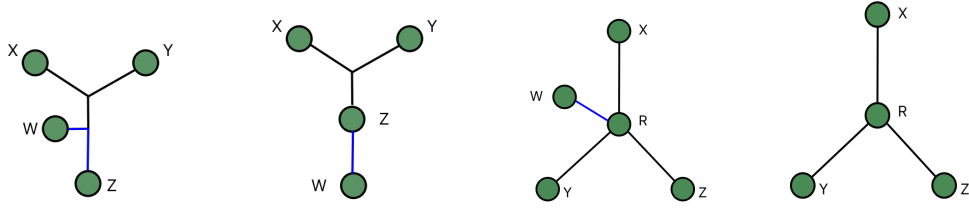

(a) $\hat{T}$ when $\pi x = z$ and $(z,x)_w = (z,y)_w < d(w,z)$.
(b) $\hat{T}$ when $\pi x = z$ and $(z,x)_w = (z,y)_w = d(w,z)$.
(c) $\hat{T}$ when $(y,x)_w = (y,z)_w = (x,z)_w \neq 0$
(d) Universal Tree on $x, y, z$.

Figure 1: Figures showing the tree $\hat{T}$ from Lemma 2 for $Zone_2(z)$ (a), $Zone_1(z)$ (b), $Zone_1(r)$ (c), and the Universal tree (d).

**Theorem 1.** *Given $(X, d)$, a $\delta$-hyperbolic metric space, and $n$ points $x_1, \ldots, x_n \in X$, TREEREP returns a tree $(T, d_T)$. In the case that $\delta = 0$, $d_T = d$, and $T$ has the fewest possible nodes. TREEREP has a worst case run time $O(n^2)$. Furthermore the algorithm is embarrassingly parallelizable.*

**Remark 1.** *In practice, we see that the run time for TREEREP is much faster than $O(n^2)$.*

To better understand the geometric insights used to develop TREEREP, we first focus on the problem of reconstructing the tree structure from a tree metric. Algorithm 1 and 2 present a high level version of the pseudo-code. The complete pseudo-code for TREEREP is presented in Appendix 13.

TREEREP is a recursive, divide and conquer algorithm. The first main idea (Lemma 1) is that for any metric $d$ on three points, we can construct a tree $T$ with four nodes such that $d_T = d$. We will call such trees *universal trees*. The second main idea (Lemma 2) is that adding a fourth point to a universal tree can be done consistently and, more importantly, the additional point falls into one of seven different zones. Thus, TREEREP will first create a universal tree $T$ and then will sort the remaining data points into the seven different zones. We will then do recursion with each of the zones.

**Lemma 1.** *Given a metric $d$ on three points $x, y, z$, there exists a (weighted) tree $(T, d_T)$ on four nodes $x, y, z, r$, such that $r$ is adjacent to $x, y, z$, the edge weights are given by $d_T(x, r) = (y, z)_x$, $d_T(y, r) = (x, z)_y$ and $d_T(z, r) = (x, y)_z$, and the metric $d_T$ on the tree agrees with $d$.*

**Definition 9.** *The tree constructed in Lemma 1 is the universal tree on the three points $x, y, z$. The additional node $r$ is known as a Steiner node.*

An example of the universal tree can be seen in Figure 1(d). To understand the distinction between the seven different zones, we need to reinterpret Equation 2.1. We know that for any tree metric, and any four points $w, x, y, z$, we have that

$$(x, y)_w \geq \min\big((x, z)_w, (y, z)_w\big).$$

This inequality implies that the smaller two of the three numbers $(x, y)_w$, $(x, z)_w$, and $(y, z)_w$ are equal. In this case, knowing which of the quantities are equal tells us the structure of the tree. Specifically, here $x, y, z$ will be the three points in our universal tree $T$ and $w$ will be the point that we want to sort. Then initially, we have four possibilities. The first possibility is that all three Gromov products are equal. This case will define its own zone. If this is not the case, then we have three possibilities depending on which two out of the three Gromov products are equal. Suppose we have that $(x, y)_w = (x, z)_w$, then due to the triangle inequality, we have that $d(w, x) \geq (x, y)_w$. Thus, we will further subdivide this case into two more cases, depending on whether $d(w, x) = (x, y)_w$ or $d(w, x) > (x, y)_w$. Each of these cases will define their own zone. Examples of the different cases can be seen in Figure 1. We can also see that there are two different types of zones. The first type is when we connect the new node directly to an existing node as seen in Figures 1(b) and 1(c). The second type is when we connect $w$ to an edge as seen in Figure 1(a). The formal definitions for the zones can be seen in Definition 13.

**Lemma 2.** *Let $(X, d)$ be a tree space. Let $w, x, y, z$ be four points in $X$ and let $(T, d_T)$ be the universal tree on $x, y, z$ with node $r$ as the Steiner node. Then we can extend $(T, d_T)$ to $(\hat{T}, d_{\hat{T}})$ to include $w$ such that $d_{\hat{T}} = d$.*

**Definition 10.** *Given a data set $V$ (consisting of data points, along with the distances amongst the points), a universal tree $T$ on $x, y, z \in V$ (with $r$ as the Steiner node), let us define the following two zone types.*

1. *The definition for zones of type one is split into the following two cases.*
    (a) *$Zone_1(r) = \{w \in V : (x, y)_w = (y, z)_w = (z, x)_w\}$*
    (b) *For a given permutation $\pi$ on $\{x, y, z\}$, $Zone_1(\pi x) = \{w \in V : (\pi x, \pi y)_w = (\pi x, \pi z)_w = d(w, \pi x)\}$*
2. *For a given permutation $\pi$ on $\{x, y, z\}$, $Zone_2(\pi x) = \{w \in V : (\pi x, \pi y)_w = (\pi x, \pi z)_w < d(w, \pi x)\}$*

Using this terminology and our structural lemmas, we can describe a recursive algorithm that reconstructs the tree structure from a 0-hyperbolic metric. Given a data set $V$ we pick three random points $x, y, z$ and construct the universal tree $T$. Then for all other $w \in V$, sort the $w$'s into their respective zones. Then for each of the seven zones we can recursively build new universal trees. For zones of type 1, pick any two points, $w_{i_1}, w_{i_2}$ and form the universal tree for $\pi x$ (or $r$), $w_{i_1}, w_{i_2}$. If there is only one node in this zone, connect it to $\pi x$ (or $r$). For zones of type 2, pick any one point, $w_{i_1}$ and form the universal tree for $\pi x, w_{i_1}, r$. Note that during the recursive step for zones of type 2, we create universal trees with Steiner nodes $r$ as one of the nodes. Hence we need to compute the distance from $r$ to all other nodes sent to that zone. We can calculate this when we first place $r$. Concretely, if $r$ is the Steiner node for the universal tree $T$ on $x, y, z$, then for any $w$, we will have that $d(w, r) = \max((x, y)_z, (y, z)_x, (z, x)_y)$. The proof for the consistency of this formula is in the proof of Lemma 2.

Finally, to complete the analysis, the following lemma proves that we only need to check consistency of the metric within each zone to ensure global consistency.

**Lemma 3.** *Given $(X, d)$ a metric tree, and a universal tree $T$ on $x, y, z$, we have the following*

1. *If $w \in Zone_1(x)$, then for all $\hat{w} \notin Zone_1(x)$, we have that $x \in g(w, \hat{w})$.*
2. *If $w \in Zone_2(x)$, then for all $\hat{w} \notin Zone_i(x)$ for $i = 1, 2$, we have that $r \in g(w, \hat{w})$.*

**TreeRep for General $\delta$-Hyperbolic Metrics.** Having seen the main geometric ideas behind TREEREP, we want to extend the algorithm to return an approximating tree for any given metric. For an arbitrary $\delta$-hyperbolic metric, Lemma 2 does not hold. We can, however, modify it and leverage the intuition behind the original proof. Given four points $w, x, y, z$, we do not satisfy one of the conditions of Lemma 2, if all three Gromov products $(x, y)_w, (x, z)_w, (y, z)_w$ have distinct values. Nevertheless, we can still compute the maximum of these three quantities. Furthermore, since we have a $\delta$-hyperbolic metric, the smaller two products will be within $\delta$ of each other. Let us suppose that $(x, y)_w$ is the biggest. Then we place $w$ in $Zone_1(x)$ if and only if $d(z, w) = (y, z)_w$ or $d(z, w) = (x, z)_w$. Otherwise we place $w \in Zone_2(x)$. Note that when we have tree metric, we have that $d(z, w) = (y, z)_w$ if and only if $d(z, w) = (x, z)_w$.

As shown by Proposition 1, when we do this, we are introducing a distortion of at most $\delta$ between $w$ and $y, z$. This suggests that when we do zone 2 recursive steps, we should pick the node that closest to $r$ as the third node for the universal tree. We see experimentally that this significantly improves the quality of the tree returned. Note, we do not have a global distortion bound for when the input is a general $\delta$-hyperbolic metric. However, as we will see experimentally, we tend to produce trees with low distortion.

**Proposition 1.** *Given a $\delta$-hyperbolic metric $d$, the universal tree $T$ on $x, y, z$ and a fourth point $w$, when sorting $w$ into its zone ($zone_i(\pi x)$), TREEREP introduces an additive distortion of at most $\delta$ between $w$ and $\pi y, \pi z$.*

**Steiner nodes.** A Steiner node is any node that did not exist in the original graph that one adds to it. We give a simple example to illustrate that Steiner nodes are necessary for reconstructing the correct tree. Additionally, we demonstrate that forming a graph and then computing any spanning tree (as done in [2, 16, 33]) will not recover the tree structure. Consider 3 points $x, y, z$ such that all pairwise distances are equal to 2. Then, the associated graph is a triangle and any spanning tree

---
**Algorithm 1** Metric to tree structure algorithm.
___
1: **function** TREE STRUCTURE(X, $d$)
2:      $T = (V, E, d') = \emptyset$
3:      Pick any three data points uniformly at random $x, y, z \in X$.
4:      $T$ = RECURSIVE_STEP($T, X, x, y, z, d, d_T$,)
5:      **return** $T$
6: **function** RECURSIVE_STEP($T, X, x, y, z, d, d_T$,)
7:      Construct universal tree for $x, y, z$ and sort the other nodes into the seven zones.
8:      Recurse for each of the seven zones by calling ZONE1_RECURSION and ZONE2_RECURSION. **return** $T$
___

---
**Algorithm 2** Recursive parts of TreeRep.
___
1: **function** ZONE1_RECURSION($T, d_T, d, L, v$)
2:      **if** Length($L$) == 0 **then return** $T$
3:      **if** Length($L$) == 1 **then**
4:          Let $u$ be the one element in $L$ and add edge $(u, v)$ to $E$ with weight $d_T(u, v) = d(u, v)$
5:          **return** $T$
6:      Pick any two $u, z$ from $L$ and remove them from $L$
7:      **return** RECURSIVE_STEP($T, L, v, u, z, d, d_T$)
8: **function** ZONE2_RECURSION($T, d_T, d, L, u, v$)
9:      **if** Length($L$) == 0 **then return** $T$
10:      Set $z$ to be the closest node to $v$ and delete edge $(u, v)$
11:      **return:** RECURSIVE_STEP($T, L, v, u, z, d, d_T$)
___

is a path. Then, the distance between the endpoints of the spanning tree is not correct; it has been distorted or stretched. The "correct" tree is obtained by adding a new node $r$ and connecting $x, y, z$ to $r$, and making all the edge weights equal to 1. Thus, we need Steiner nodes when reconstructing the tree structure. Methods such as MST and LS [2] that do not add Steiner nodes will not produce the correct tree, when given a 0-hyperbolic metric, even though such algorithms do come with upper bounds on the distortion of the distances. In this setting, we want to obtain a tree that as accurately as possible represents the metric even at the cost of additional nodes; we do not simply want a tree that is a subgraph of a given graph.

## 4    Experiments

In this section, we demonstrate the effectiveness of TREEREP. Additional details about the experiments and algorithms can be found in Appendix 12.[5]

For the first task of approximating metrics with tree metrics, we compare TREEREP against algorithms that find approximating trees; Minimum Spanning Trees (MST) [33], LEVELTREES (LT) [11], NEIGHBOR JOIN (NJ) [35], Low Stretch Trees (LS) [2, 16], CONSTRUCTTREE (CT) [1], and PROBTREE (BT) [3]. When comparing against such methods, we show that not only is TREEREP much faster than all of the above algorithms (except MST, and LS), but that TREEREP produces better quality metrics than MST, LS, LT, BT, and CT and metrics that are competitive with NJ. In addition to these methods, other methods such as UPGMA [39] also learn tree structures. However, these algorithms have other assumptions on the data. In particular, for UPGMA, the additional assumption is that the metric is an ultrametric. Hence we do not compare against such methods.

One important distinction between methods such as LS, MST, and LT and the rest, is that LS, MST, and LT require a graph as the input. This graph is crucial for these methods and hence sets these methods apart from the rest, as the rest only require a metric.

For the second task of learning hyperbolic embeddings, we compare TREEREP against Poincare Maps (PM) [30], Lorentz Maps (LM) [31], PT [36], and hMDS [36]. Since we can embed trees into $\mathbb{H}^k$ with arbitrarily low distortion, we think of trees as hyperbolic representations. When comparing

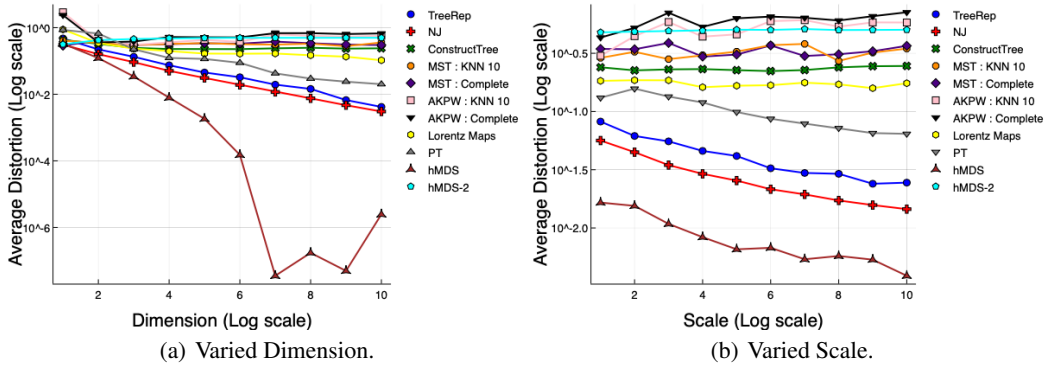

(a) Varied Dimension.　　　　　　　　　　　　(b) Varied Scale.

Figure 2: Average distortion of the metric learned for 100 randomly sampled points from $\mathbb{H}^k$ for $k = 2^i$ and from $\mathbb{H}^{10}$ for scale $s = 2^i$ for $i = 1, 2, \ldots, 10$.

against such methods, we show that TREEREP is not only four to five orders of magnitude faster, but for low dimensions, and in many high dimensional cases, produces better quality embeddings.

We first perform a benchmark test for tree reconstruction from tree metrics. Then, for both tasks, we test the algorithms on three different types of data sets. First, we create synthetic data sets by sampling random points from $\mathbb{H}^k$. Second, we will take real world biological data sets that are believed to have hierarchical structure. Third, we consider metrics that come from real world unweighted graphs. In each case, we will show that TREEREP is an extremely fast algorithm that produces as good or better quality metrics. We will evaluate the methods on the basis of computational time, and the average distortion, as well as mean average precision (MAP) of the learned metrics.[6]

**Remark 2.** TREEREP *is a randomized algorithm, so all numbers reported are averaged over 20 runs. The best number produced by* TREEREP *can be found in the Appendix.*

**Tree Reconstruction Experiments.** Before experimenting with general $\delta$-hyperbolic metrics, we benchmark our method on 0-hyperbolic metrics. To do this, we generate random synthetic 0-hyperbolic metrics. More details can be found in Appendix 12. Since TREEREP and NJ are the only algorithms that are theoretically guaranteed to return a tree that is consistent with the original metric, we will run this experiment with these two algorithms only. We compare the two algorithms based on their running times and the number of nodes in the trees. As we can see from Table 1, TREEREP is a much more viable algorithm at large scales. Additionally, the trees returned by NJ have double the number of nodes as the original trees. Contrarily, the trees returned by TREEREP have exactly the same number of nodes as the original trees.

Table 1: Time taken by Nj and TreeRep to reconstruct the tree structure.

| n | 11 | 40 | 89 | 191 | 362 | 817 | 1611 |
|----|-------|--------|--------|--------|------|-----|------|
| TR | 0.053 | 0.23 | 0.0017 | 0.0039 | 0.02 | 0.08 | 0.12 |
| NJ | 0.084 | 0.0016 | 0.0067 | 0.036 | 0.18 | 1.7 | 15 |

Table 2: Time taken by PT, LM, hMDS, to learn a 10 dimensional embedding for the synthetic data sets and average time taken by TREEREP (TR), MST, and CT.

| | TR | NJ | MST | LS | CT | PT | LM | hMDS | hMDS-2 |
|------|-------|------|--------|-------|-------|-----|-----|------|--------|
| Time | 0.002 | 0.06 | 0.0001 | 0.002 | 0.076 | 312 | 971 | 11.7 | 0.008 |

**Random points on Hyperbolic Manifold.** We generate two different types of data sets. First, we hold the dimension $k$ constant and scale the coordinates. Second, we hold the magnitude of the

coordinates constant and increase the dimension $k$. Note these metrics do not come with an underlying graph! Hence to even apply methods such as MST, or LS we need to do some work. Hence, we create two different weighted graphs; a complete graph and a nearest neighbor graph.

For both types of data, Figures 2(a) and 2(b) show that as the scale and the dimension increase, the quality of the trees produced by TREEREP and NJ get better. Contrastingly, the quality of the trees produced by MST, CONSTRUCTTREE, and LS do not improve. Hence we see that when we do not have an underlying sparse graph that was used to generate the metric, methods such as MST and LS do not perform well. In fact, they have the worst performance. This greater generality of possible inputs is one of the major advantages of our method. Thus, demonstrating that TREEREP is an extremely fast algorithm that produces good quality trees that approximate hyperbolic metrics. Furthermore, Table 2 shows that TREEREP is a much faster algorithm than NJ.

For the second task of finding hyperbolic embeddings, we compare against LM, PT and hMDS. For both LM, PT, and hMDS, we compute an embedding into $\mathbb{H}^k$, where $k$ is dimension of the manifold the data was sampled from. We also use hMDS to embed into $\mathbb{H}^2$, we call this hMDS-2. We can see from Figures 2(a) and 2(b) that TREEREP produces *much better* embeddings than LM, PT, and hMDS-2. Furthermore, LM and PT are extremely slow, with PT and LM taking 312 and 917 seconds on average, respectively. Thus, showing that TREEREP is 5 orders of magnitude faster than LM and PT, *and produces better quality representations.* On the other hand, since our points come from $\mathbb{H}^k$ if we try embedding into $\mathbb{H}^k$ with hMDS we should theoretically have zero error. However, these are high dimensional representations. We want low dimensional hyperbolic representations. Thus, we compared against hMDS-2 which did not perform well.

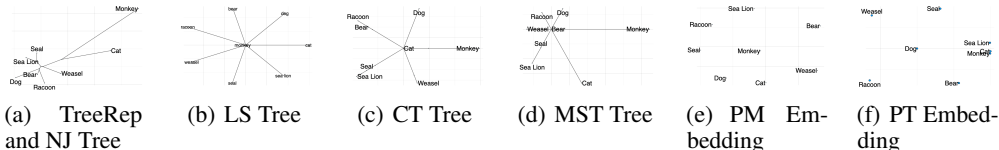

| (a) TreeRep and NJ Tree | (b) LS Tree | (c) CT Tree | (d) MST Tree | (e) PM Embedding | (f) PT Embedding |

Figure 3: Tree structure and embeddings for the Immunological distances from [37].

Table 3: Time taken in seconds and the average distortion of the tree metric learned by TREEREP, NJ, MST, and CT and of the 2-dimensional hyperbolic representation learned by PM and PT on the Zeisel and CBMC data set. The numbers for TREEREP (TR) are the average numbers over 20 trials.

| | Zeisel | | | | | | | CBMC | | | |
|---|---|---|---|---|---|---|---|---|---|---|---|
| | TR | NJ | MST | LS | CT | PT | PM | TR | NJ | MST | LS |
| Time | 0.36 | 122.2 | 0.11 | 7.2 | >14400 | 8507 | 12342 | 2.8 | >14400 | 0.55 | 30 |
| Distortion | 0.117 | 0.144 | 0.365 | 0.250 | n/a | 0.531 | 0.294 | 0.260 | n/a | 1.09 | 1.45 |

**Biological Data: scRNA seq and phylogenetic data.** We also test on three real world biological data sets. The first data set consists of immunological distances from Sarich [37]. Given these distances, the goal is to recover the hierarchical phylogenetic structure. As seen in Figure 3, the trees returned by TREEREP and NJ recover this structure well, with sea lion and seal close to each other, and monkey and cat far away from everything else. Divergently, the trees and embeddings produced by MST, LS, CONSTRUCTTREE, PM, and PT make less sense as phylogenetic trees.

The second type of data sets are the Zeisel and CBMC sc RNA-seq data set [44, 40]. These data sets are expected to be a tree as demonstrated in Dumitrascu et al. [14]. Here we used the various algorithms to learn a tree structure on the data or to learn an embedding into $\mathbb{H}^2$. The time taken and the average distortion are reported in Table 3. In this case, we see that TREEREP has the *lowest* distortion. Additionally, TREEREP is 20 times faster than NJ and is 20,000 to 40,000 times faster than PT and PM. Furthermore, NJ, CT, PT, and PM timed out (took greater than 4 hours) on the CBMC data set. For the CBMC data set, we see that TREEREP is *only* algorithm that produces good quality embeddings in a reasonable time frame. Again we see that if the input is a metric instead of a graph, algorithms such as MST and LS do not do well. We also tried to use hMDS for this experiment, but it either didn't output a metric or it outputted the all zero metric.

Table 4: Table with the time taken in seconds, MAP, and average distortion for all of the algorithms when given metrics that come from unweighted graph. Darker cell colors indicates better numbers for MAP and average distortion. The number next to PT, PM, LM is the dimension of the space used to learn the embedding. The numbers for TREEREP (TR) are the average numbers over 20 trials.

| Graph | | TR | NJ | MST | LT | CT | LS | PT 2 | PT 200 | PM 2 | LM 2 | LM 200 | PM 200 |
|---|---|---|---|---|---|---|---|---|---|---|---|---|---|
| | $n$ | | | | | | MAP | | | | | | |
| Celegan | 452 | 0.473 | 0.713 | 0.337 | 0.272 | 0.447 | 0.313 | 0.098 | 0.857 | 0.479 | 0.466 | 0.646 | 0.662 |
| Dieseasome | 516 | 0.895 | 0.962 | 0.789 | 0.725 | 0.815 | 0.785 | 0.392 | 0.868 | 0.799 | 0.781 | 0.874 | 0.886 |
| CS Phd | 1025 | 0.979 | 0.993 | 0.991 | 0.964 | 0.807 | 0.991 | 0.190 | 0.556 | 0.537 | 0.537 | 0.593 | 0.593 |
| Yeast | 1458 | 0.815 | 0.892 | 0.871 | 0.742 | 0.859 | 0.873 | 0.235 | 0.658 | 0.522 | 0.513 | 0.641 | 0.643 |
| Grid-worm | 3337 | 0.707 | 0.800 | 0.768 | 0.657 | - | 0.766 | - | - | 0.334 | 0.306 | 0.558 | 0.553 |
| GRQC | 4158 | 0.685 | 0.862 | 0.686 | 0.480 | - | 0.684 | - | - | 0.589 | 0.603 | 0.783 | 0.784 |
| Enron | 33695 | 0.570 | - | 0.524 | - | - | 0.523 | - | - | - | - | - | - |
| Wordnet | 74374 | 0.984 | - | 0.989 | - | - | 0.989 | - | - | - | - | - | - |
| | $m$ | | | | | | Average Distortion | | | | | | |
| Celegan | 2024 | 0.197 | 0.124 | 0.255 | 0.166 | 0.325 | 0.353 | 0.236 | 0.096 | 0.236 | 0.249 | 0.224 | 0.211 |
| Dieseasome | 1188 | 0.188 | 0.161 | 0.161 | 0.157 | 0.315 | 0.228 | 0.227 | 0.05 | 0.323 | 0.328 | 0.335 | 0.332 |
| CS Phd | 1043 | 0.204 | 0.134 | 0.298 | 0.161 | 0.282 | 0.291 | 0.295 | 0.105 | 0.374 | 0.378 | 0.378 | 0.380 |
| Yeast | 1948 | 0.205 | 0.149 | 0.243 | 0.243 | 0.282 | 0.243 | 0.230 | 0.089 | 0.246 | 0.248 | 0.234 | 0.234 |
| Grid-worm | 6421 | 0.188 | 0.135 | 0.171 | 0.202 | - | 0.234 | - | - | 0.196 | 0.203 | 0.192 | 0.193 |
| GRQC | 13422 | 0.192 | 0.200 | 0.275 | 0.267 | - | 0.206 | - | - | 0.212 | 0.198 | 0.193 | 0.193 |
| Enron | 180810 | 0.453 | - | 0.607 | - | - | 0.562 | - | - | - | - | - | - |
| Wordnet | 75834 | 0.131 | - | 0.336 | - | - | 0.071 | - | - | - | - | - | - |
| | $\delta$ | | | | | | Time in seconds | | | | | | |
| Celegan | 0.21 | 0.014 | 0.28 | 0.0002 | 0.086 | 0.9 | 0.001 | 573 | 1156 | 712 | 523 | 1578 | 1927 |
| Dieseasome | 0.17 | 0.017 | 0.41 | 0.0003 | 0.39 | 15.76 | 0.001 | 678 | 1479 | 414 | 365 | 978 | 1112 |
| CS Phd | 0.23 | 0.037 | 2.94 | 0.0007 | 1.97 | 226 | 0.006 | 1607 | 4145 | 467 | 324 | 768 | 1149 |
| Yeast | $\leq 0.32$ | 0.057 | 8.04 | 0.0008 | 8.21 | 957 | 0.001 | 9526 | 17876 | 972 | 619 | 1334 | 2269 |
| Grid-worm | $\leq 0.38$ | 0.731 | 163 | 0.001 | 191 | - | 0.007 | - | - | 2645 | 1973 | 4674 | 5593 |
| GRQC | $\leq 0.36$ | 0.42 | 311 | 0.0014 | 70.9 | - | 0.006 | - | - | 7524 | 7217 | 9767 | 1187 |
| Enron | - | | 27 | - | 0.013 | - | - | 0.13 | - | - | - | - | - | - |
| Wordnet | - | | 74 | - | 0.18 | - | - | 0.08 | - | - | - | - | - | - |

**Unweighted Graphs.** Finally, we consider metrics that come from unweighted graphs. We use eight well known graph data sets from [34]. Table 7 records the performance of all the algorithms for each of these data sets. For learning tree metrics to approximate general metrics, we see that NJ has the best MAP, with TREEREP, MST, and LS tied for second place. In terms of distortion, NJ is the best, TREEREP is second, while MST is third and LS is sixth. However, NJ is extremely slow and is not viable at scale. Hence, in this case, we have three algorithms with good performance at large scale; TREEREP, MST, and LS. However, MST and LS did not perform well in the previous experiments.

For the task of learning hyperbolic representations, we see that PM, LM, and PT are much slower than the methods that learn a tree first. In fact, these algorithms were too slow to compute the hyperbolic embeddings for the larger data sets. Additionally, this extra computational effort does not always result in improved quality. In all cases, except for the Celegan data set, the MAP returned by TREEREP is superior to the MAP of the 2-dimensional embeddings produced by PM, LM, and PT. In fact, in most cases, these 2-dimensional embeddings, have worse MAP than all of the tree first methods. Even when they learn 200-dimensional embeddings, PM, LM and PT have worse MAP than TREEREP on most of the data sets. Furthermore, except for PT200, the average distortion of the metric returned by TREEREP is superior to PT2, PM, an LM. Thus, showing the effectiveness of TREEREP at learning good Hyperbolic representations quickly.

# 5 Broader Impact

There are multiple aspects to the broader impacts of our work, from the impact upon computational biology, specifically, to the impact upon data sciences more generally. The potential impacts on society, both positive and negative, are large. Computational biology is undergoing a revolution due to simultaneous advances in the creation of novel technologies for the collection of multiple and novel sources of data, and in the progress of the development of machine learning algorithms for the analysis of such data. Social science has a similar revolution in its use of computational techniques for the analysis and gathering of data.

Cellular differentiation is the process by which cells transition from one cell type (typically an immature cell) into more specialized types. Understanding how cells differentiate is a critical problem in modern developmental and cancer biology. Single-cell measurement technologies, such as single-cell RNA-sequencing (scRNA-seq) and mass cytometry, have enabled the study of these processes. To visualize, cluster, and infer temporal properties of the developmental trajectory, many researchers have developed algorithms that leverage hierarchical representations of single cell data. To discover these geometric relationships, many state-of-the-art methods rely on distances in low-dimensional Euclidean embeddings of cell measurements. This approach is limited, however, because these types of embeddings lead to substantial distortions in the visualization, clustering, and the identification of cell type lineages. Our work is specifically focused on extracting and representing hierarchical information.

On the more negative side, these algorithms might also be used to analyze social hierarchies and to divine social structure from data about peoples' interactions. Such tools might encourage, even justify, the intrusive and pervasive collection of data about how people interact and with whom.

*Work partially supported by funds from the Michigan Institute for Data Science.*

## Footnotes

[2]A similar idea is mentioned in Sala et al. [36] for graph inputs rather than general metrics. They do not, however undertake a detailed exploration of the idea.

[3]Note that the input to our problem are metrics and not graphs. Thus, we handle more general inputs as compared to Alon et al. [2], Elkin et al. [16], Chepoi et al. [11], and Prim [33].

[4]Note, such metrics may have representations as graphs that are not trees, Section 3 has a simple example.

[5]All code can be found at the following link `https://github.com/rsonthal/TreeRep`

[6]MAP is used in [30, 31, 36], while average distortion is used in [36]. The definitions are in the appendix.

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
