[Supplementary Material 1 · Full.pdf]

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

[7]For NJ and LT, computing this $\alpha$ made the average distortion worse, so we report numbers un-scaled. Additionally, computing $\alpha$ is too computationally expensive for bigger data sets and was not done for the Enron and Wordnet data set.

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

# 6 Metric First Discussion and Justification

Table 7 shows that for most of the data sets, learning a tree structure first and then embedding it into hyperbolic space, yields embeddings with better MAP and average distortion compared to methods that learn the embedding directly. One possible explanation for this phenomenon is that the optimization problems that seek the embeddings directly are not being solved optimally. That is, the algorithms get stuck at some local minimum. Another possibility is that there is a disconnect between the objective being optimized and the statistics calculated to judge the quality of the embeddings.

We propose that there are geometric facts about hyperbolic space that suggest embedding by first learning a tree is the correct approach. The tree-likeness of hyperbolic space has been studied from many different approaches. We present details from Hamann [22], Dyubina and Polterovich [15] and looks at the geometry of $\mathbb{H}^k$ at its two extremes; large scale and small scale. Since $\mathbb{H}^k$ is a manifold, we know that at small scales hyperbolic space looks like Euclidean space. Additionally, in the Poincare disk, the hyperbolic Riemannian metric is given by $\frac{4}{(1-x^2-y^2)^2}(dx^2 + dy^2)$ and is just a re-scaling of the Euclidean metric. Thus, at small scales, hyperbolic space is similar to Euclidean space.

Hence to take advantage of hyperbolic representations (i.e., why learn a hyperbolic representation instead of a Euclidean one), we want to embed data into $\mathbb{H}^k$ at scale. To study the large scale geometry of $\mathbb{H}^k$, we consider the asymptotic cone for hyperbolic space $Con(\mathbb{H}^k)$. In particular, we can think of the asymptotic cone as the "view of our space from infinitely far away". See the more detailed discussion in Appendix 8 for examples and complete definitions. The following connects $Con(\mathbb{H}^k)$ to $\mathbb{R}$-tree spaces.

**Theorem 2.** *[42] $Con(\mathbb{H}^k)$ is a complete $\mathbb{R}$-tree.*

Thus, we see that the large scale structure of hyperbolic space is a tree, indicating a strong connection between learning trees and learning hyperbolic embeddings. Furthermore, it can be shown that $Con(\mathbb{H}^k)$ is a $2^{\aleph_0}$-universal tree. That is, any tree with finitely many nodes can be embedded into $Con(\mathbb{H}^k)$ exactly. However, these are still embeddings into $Con(\mathbb{H}^k)$. We would like to study embeddings into $\mathbb{H}^k$.

**Definition 11.** *A metric space $(T, d_T)$ admits an isometric embedding at infinity into the space $(X, d_X)$ if there exists a sequence of positive scaling factors $\lambda_i \to \infty$ such that for every point $t \in T$, there exists an infinite sequence $\{x_t^i\}, i = 1, 2, \ldots$ of points in $X$ such that for all $t_1, t_2 \in T$ $\lim_{i \to \infty} d_X(x_{t_1}^i, x_{t_2}^i)/\lambda_i = d_T(t_1, t_2)$*

**Theorem 3.** *[15] $Con(\mathbb{H}^k)$ can be isometrically embedded at infinity into $\mathbb{H}^k$.*

Thus, we can embed any tree into $\mathbb{H}^k$ with arbitrarily low distortion. A type of converse is also true.

**Definition 12.** *A (geodesic) ray $R$ is a (isometric) homeomorphic image of $[0, \infty)$, such that for any ball $B$ of finite diameter, $R$ lies outside $B$ eventually.*

Hamann [22] showed that we can construct a rooted $\mathbb{R}$-tree $T$ inside $\mathbb{H}^k$, such that every geodesic ray in $\mathbb{H}^k$ eventually converges to a ray of $T$. Thus, showing that any configuration of points at scale in $\mathbb{H}^k$ can be approximated by a tree. Additionally, larger the scale points can be better approximated by trees. More details can be found in Appendix 9. Thus, showing that learning a tree and then embedding this tree into $\mathbb{H}^k$ is equivalent to learning hyperbolic representations at scale.

This provides an explanation for why as the scale and dimension increased, TREEREP found a tree that better approximated the hyperbolic metric in Section 4. This also provides a justification for why learning a tree first, results in better hyperbolic representations.

# 7 Proofs

## 7.1 Tree Representation Proofs

**Lemma 1.** *Given a metric $d$ on three points $x, y, z$, there exists a (weighted) tree $(T, d_T)$ on four nodes $x, y, z, r$, such that $r$ is adjacent to $x, y, z$, the edge weights are given by $w(x, r) = (y, z)_x$, $w(y, r) = (x, z)_y$ and $w(z, r) = (x, y)_z$, and the metric $d_T$ on the tree agrees with $d$.*

*Proof.* The basic structure of this tree can be seen in Figure 1(d). To prove that the metrics agree we such need to see the following calculation.

$$
\begin{aligned}
d_T(x,y) &= w(x,r) + w(r,y) \\
&= (y,z)_x + (x,z)_y \\
&= \frac{1}{2}(d(x,y) + d(x,z) - d(y,z) \\
&\quad + d(x,y) + d(y,z) - d(x,z)) \\
&= d(x,y)
\end{aligned}
$$

Here $d_T$ is the metric on the tree $T$. $\qquad\square$

One important fact that we need is that if $(X,d)$ is a metric graph, then for any three distinct points $x, y, z \in X$, the geodesics connecting them intersect at a unique point. As seen in Lemma 1, we refer to this point a Steiner point $r$. It is now important to note that even though $r$ may not be a point in the data set we are given, but $r \in X$ [7]. Thus, in the following lemmas, whenever we find a Steiner point, we will assume that the metric $d$ is defined on $r$.

**Lemma 4.** *If $d$ is a tree metric and $x, y, w$ are three points then*

1. $(x,y)_w = 0$ *if and only if* $w \in g(x,y)$
2. $(x,y)_w = d(x,w)$ *if and only if* $(w,y)_x = 0$.
3. $(x,y)_w = d(y,w)$ *if and only if* $(w,x)_y = 0$.

Here $g(x,y)$ is the unique path connecting $x$ and $y$.

*Proof.* For 1. we see that

$$
\begin{aligned}
0 = (x,y)_w &= \frac{1}{2}(d(w,x) + d(w,y) - d(x,y)) \\
&\Rightarrow d(x,y) = d(w,x) + d(w,y)
\end{aligned}
$$

Thus we have that $w \in g(x,y)$.

For 2. we see that

$$
\begin{aligned}
(x,y)_w = d(x,w) &\Rightarrow d(w,x) + d(w,y) - d(x,y) \\
&= 2d(w,x) \\
&\Rightarrow d(w,x) + d(x,y) - d(w,y) = 0 \\
&\Rightarrow 2(w,y)_x = 0
\end{aligned}
$$

The proof for 3 is similar to that of 2.

$\qquad\square$

**Lemma 2.** *Let $(X,d)$ be a tree space. Let $w, x, y, z$ be four points in $X$ and let $(T, d_T)$ be the universal tree on $x, y, z$ with node $r$ as the Steiner node. Then we can extend $(T, d_T)$ to $(\hat{T}, d_{\hat{T}})$ to include $w$ such that $d_{\hat{T}} = d$.*

*Proof.* We note that there are four different possible cases for the configuration of $x, y, z, w$ depending on the relationship amongst the Gromov products. Each case determines a different placement of $r$, as follows:

1. If $(x,y)_w = (x,z)_w = (y,z)_w = 0$, then replace $r$ with $w$ to obtain $\hat{T}$.
2. If $(x,y)_w = (x,z)_w = (y,z)_w = c > 0$, then connect $w$ to $r$ via an edge of weight $c$ to obtain $\hat{T}$.
3. If there exists a permutation $\pi : \{x, y, z\} \to \{x, y, z\}$ such that,

$$
(\pi x, \pi y)_w = (\pi x, \pi z)_w = c < (\pi y, \pi z)_w
$$

and $d(\pi x, w) = (\pi x, \pi y)_w$, then connect $w$ to $\pi x$ via an edge of weight $c$ to obtain $\hat{T}$.

4. If there exists a permutation $\pi : \{x, y, z\} \to \{x, y, z\}$ such that,

$$(\pi x, \pi y)_w = (\pi x, \pi z)_w = c < (\pi y, \pi z)_w$$

and $d(\pi x, w) > (\pi x, \pi y)_w$, then add a Steiner point $\hat{r}$ on the edge $x, r$ with $d(\pi x, \hat{r}) = d(\pi x, w) - c$ and connect $w$ to $\hat{r}$ via an edge of weight $c$ to obtain $\hat{T}$.

To prove that these extensions of $T$ are consistent, first let us prove that there are exactly four cases. To do that, first note that since we have a $0$-hyperbolic metric, at least two of the three Gromov products must be equal. Using the triangle inequality, we can see that for any three points $a, b, c$ the following holds

$$0 \le (a, b)_c \le d(a, c).$$

That is, either we are in the first two cases and three of products are equal, or we have that two of the products are equal. In the case that two of the products are equal, the permutation $\pi$ tells us which of the two are equal and we further subdivide into the case whether $d(\pi x, w) = (\pi x, \pi y)_w$ or $d(\pi x, w) > (\pi x, \pi y)_w$ as we cannot have $d(\pi x, w) < (\pi x, \pi y)_w$.

Therefore, there are at most four possible configuration cases and it remains to show that the new tree $d_{\hat{T}}$ is consistent with $d$ on the four points. In each case, we present the high level intuition for why these modification result in a consistent tree. The low level details about the metric numbers can easily be checked.

**Case 1:** If $(x, y)_w = (x, z)_w = (y, z)_w = 0$, then we replaced $r$ with $w$ in $\hat{T}$. In this case, using Lemma 4, we see that $w$ must lie on all tree geodesics $g(x, y), g(x, z), g(y, z)$. Since the metric comes from a tree, these three geodesics can only intersect at one point $r$. Thus, we must replace $r$ with $w$.

To see that the metric is consistent, we need to verify that $d(w, x) = d_{\hat{T}}(r, x)$. To see we have the following:

$$\begin{aligned}
d_{\hat{T}}(r, x) &= (y, z)_x \\
&= (y, z)_x + (x, y)_w + (x, z)_w - (y, z)_w \\
&= d(w, x)
\end{aligned}$$

**Case 2:** If

$$(x, y)_w = (x, z)_w = (y, z)_w = c > 0,$$

then we can see that $(x, w)_r = (y, w)_r = (z, w)_r = 0$. In this case, $r$ lies on geodesics $g(x, y)$, $g(x, z), g(x, w), g(y, w), g(y, z), g(z, w)$. Thus, we must have a star shaped graph with $r$ in the center.

To see that the metric is consistent we just need to verify that $d(w, x) = d_{\hat{T}}(w, x)$. To see that we have the following calculation.

$$\begin{aligned}
d_{\hat{T}}(w, x) &= d_{\hat{T}}(w, r) + d_{\hat{T}}(x, r) \\
&= (x, y)_w + (y, z)_x \\
&= (x, y)_w + (x, z)_w - (y, z)_w + (y, z)_x \\
&= d(w, x)
\end{aligned}$$

**Case 3:** In this case suppose condition 4 is true. Without loss of generality assume that $\pi$ is the identity map. In each case, we have a tree that looks like a tree in Figure 1. In this case, we can do the calculations and see that $(w, y)_r = (w, z)_r = 0$. That is, the geodesics $g(w, y), g(w, z), g(y, z), g(x, y), g(x, z)$ all intersect at the same point. Thus, again telling us our tree structure.

To check that the metric is consistent, we need to verify that $d(w, y) = d_{\hat{T}}(w, y) = d_{\hat{T}}(w, r) + d_{\hat{T}}(r, y)$. Before we can do that, let us first verify that

$$d_{\hat{T}}(w, r) = (y, z)_w$$

To verify this we need to the following calculation

$$
\begin{aligned}
d_{\hat{T}}(w, r) &= d_{\hat{T}}(r, \hat{r}) + d_{\hat{T}}(\hat{r}, w) \\
&= c + d_T(x, r) - d_{\hat{T}}(x, \hat{r}) \\
&= c + (y, z)_x - (d(x, w) - c) \\
&= 2c + (y, z)_x - d(x, w) \\
&= (x, y)_w + (x, z)_w + (y, z)_x - d(w, x) \\
&= (y, z)_w
\end{aligned}
$$

We then can see that

$$
\begin{aligned}
d_{\hat{T}}(w, y) &= d_{\hat{T}}(w, r) + d_{\hat{T}}(r, y) \\
&= (y, z)_w + (x, z)_y \\
&= (y, z)_w + (x, y)_w - (x, z)_w + (x, z)_y \\
&= d(w, y)
\end{aligned}
$$

Note $d_{\hat{T}}(w, r) = (y, z)_w$ and the consistency of the metric implies that $d(w, r) = (y, z)_w$. Finally, we can see $(w, y)_r = 0$ as follows.

$$
\begin{aligned}
2(w, y)_r &= d(w, r) + d(r, y) - d(w, y) \\
&= (z, y)_w + (x, z)_y - d(w, y) \\
&= \frac{1}{2}(d(w, z) - d(w, y) + d(x, y) - d(x, z)) \\
&= (x, z)_w - (x, y)_w \\
&= 0
\end{aligned}
$$

Note that this also implies that $(w, x)_r > 0$.

**Case 4:** In this case, suppose condition 3 is true. Without loss of generality assume that $\pi$ is the identity map. Then in this case, we still have that $(w, y)_r = (w, z)_r = 0$, but in addition we have that $(w, y)_x = (w, z)_x = 0$. Thus, again telling us our tree structure.

In this case, to verify that the metric is consistent, we need to check that $d(w, y) = d_{\hat{T}}(w, y) = d_{\hat{T}}(w, x) + d_{\hat{T}}(x, y)$. To see this we have the following calculations.

$$
\begin{aligned}
d_{\hat{T}}(w, x) + d_{\hat{T}}(x, y) &= (x, y)_w + d(x, y) \\
&= 2(x, y)_w - (x, z)_w + d(x, y) \\
&= d(w, y) + (w, z)_x
\end{aligned}
$$

Thus, now it suffices to show that $(w, z)_x = 0$, which can be seen using the following calculations.

$$
\begin{aligned}
(x, z)_w = d(w, x) &\Rightarrow 0 = d(x, w) + d(x, z) - d(w, z) \\
&\Rightarrow (w, z)_x = 0
\end{aligned}
$$

This also implies that $(w, z)_r = 0$. $\qquad\square$

The proof of Lemma 2 shows that there are a number of ways to extend $T$ to include the new point $w$. To clarify our discussion of the extension of $T$, we introduce new terminology.

**Definition 13.** *Given a data set $V$ (consisting of data points, along with the distances amongst the points), a universal tree $T$ on $x, y, z \in V$ (with $r$ as the Steiner node), let us defining the following three zone types. The first type is associated only with the Steiner node $r$, while the other two types are defined for each of the original nodes $x, y, z$.*

  *1. $Zone_1(r)$ is all $w \in V$ such that condition 2 is true in Lemma 2.*

2. *For a given permutation $\pi$, $Zone_1(\pi x)$ is all $w \in V$ such that condition 3 is true in Lemma 2 with $\pi$.*
3. *For a given permutation $\pi$, $Zone_2(\pi x)$ is all $w \in V$, such that condition 4 is true in Lemma 2 with $\pi$.*

Note that there are seven zones total.

**Lemma 5.** *Let $(X, d)$ is a metric tree. Let $x, y \in X$ and let $r \in g(x, y)$ if and only if $X \setminus \{r\}$ has at least two disconnected components and $x, y$ are in distinct components.*

*Proof.* Suppose $r \in g(x, y)$. In metric trees, we know that there exist unique simple path between any two points. Therefore, if, after removing $r$, a path connecting $x, y$ remained (i.e., they are in the same component), then there are two simple paths connecting $x, y$ in $X$, which is not possible.

Suppose $x, y$ are in two separate components of $X \setminus \{r\}$, then because $X$ is path connected, the geodesic between $x$ and $y$ must pass through $r$. $\qquad \square$

**Lemma 3.** *Given $(X, d)$ a metric tree, and a universal tree $T$ on $x, y, z$, we have the following*

1. *If $w \in Zone_1(x)$, then for all $\hat{w} \notin Zone_1(x)$, we have that $x \in g(w, \hat{w})$.*
2. *If $w \in Zone_2(x)$, then for all $\hat{w} \notin Zone_i(x)$ for $i = 1, 2$, then we have that $r \in g(w, \hat{w})$.*

*Proof.* First let us prove statement 1. To do this, let us analyze the possible zones to which $\hat{w}$ belongs.

**Case 1:** Suppose $\hat{w} \in Zone_1(y)$ (similar for $\hat{w} \in Zone_1(z)$). Then we have that $d(\hat{w}, y) = (x, y)_{\hat{w}}$. This, implies that $(\hat{w}, x)_y = 0$. Thus, by Lemma 4, we have that $y \in g(\hat{w}, x)$. Similarly we have that $x \in g(w, y)$.

Now since $w \in Zone_1(x)$, we know that $g(x, w) \cap g(x, y) = \{x\}$. Similarly, know that $g(x, y) \cap g(y, \hat{w}) = \{y\}$. Then using Lemma 5, on removing $x$, we see that $w$ and $y$ are different connected components. Then since $x \notin g(\hat{w}, y)$, we see that $\hat{w}, y$ is in one connected component. Thus, $w, \hat{w}$ are in different components. Thus, $x \in g(w, \hat{w})$ by Lemma 5.

**Case 2:** Suppose $\hat{w} \in Zone_2(y)$ (similar for $\hat{w} \in Zone_2(z)$). Now let $r$ be the Steiner node of the universal tree on $x, y, z$. In this case we know from Lemma 2 that $r \in g(\hat{w}, x)$ and that $g(w, x) \cap g(x, r) = \{x\}$.

Now since $w \in Zone_1(x)$, we know that $g(x, w) \cap g(x, r) = \{x\}$. Similarly, know that $g(x, r) \cap g(r, \hat{w}) = \{r\}$. Then using Lemma 5, on removing $x$, we see that $w$ and $r$ are different connected components. Then since $x \notin g(\hat{w}, r)$, we see that $\hat{w}, r$ is in one connected component. Thus, $w, \hat{w}$ are in different components. Thus, $x \in g(w, \hat{w})$ by Lemma 5.

**Case 3:** $\hat{w} \in Zone_2(x)$. Let $r$ be the Steiner node for the universal tree on $x, y, z$. Now my Lemma 2, we know that $x \in g(w, r)$. Thus, again by removing $x$ and using Lemma 5, $r$ and $w$ are in different. We also have that by Lemma 2 $x \notin g(\hat{w}, r)$. Thus $r, \hat{w}$ are in the same connected component of $X \setminus \{x\}$. Thus, $w$ and $\hat{w}$ are in different connected components. Thus, by Lemma 5, $x \in g(w, \hat{w})$

Thus in all cases, we can see that $x \in g(w, \hat{w})$

Now let us prove statement 2. Without loss of generality assume that

$$\hat{w} \in Zone_i(y)$$

for $i = 1, 2$. Then from Lemma 2, we know that $r \notin g(w, x)$ and $r \notin g(\hat{w}, y)$, but $r \in g(x, y)$. Thus, using Lemma 5 on removing $r$, $x$ and $y$ and in different components and $w$ is in the same component as $x$ and $\hat{w}$ is in the same component as $y$. Thus, again using Lemma 5, we have that $r \in g(w, \hat{w})$.

$\qquad \square$

**Theorem 1.** *Given $(X, d)$, a $\delta$-hyperbolic metric space, and $n$ points $x_1, \ldots, x_n \in X$, TREEREP returns a tree $(T, d_T)$. In the case that $\delta = 0$, $d_T = d$, and $T$ has the fewest possible nodes. TREEREP has worst case run time $O(n^2)$. Furthermore the algorithm is embarrassingly parallelizable.*

*Proof.* The proof of this theorem follows directly from our structural lemmas. More precisely, we show that for $\delta = 0$, TREEREP returns a consistent metric via induction on $n$, the number of data points.

**Base Case:** The case when $n \leq 3$ is covered by Lemma 1. And, the case when $n = 4$ is covered by Lemma 2.

**Inductive Hypothesis:** Assume that for all $k \leq n$, our data set of $k$ points is consistent with a 0-hyperbolic metric $d$, then TREEREP returns a tree $(T, d_T)$ that is consistent with $d$ on the $k$ points.

**Inductive Step:** Assume that $w$ is the last vertex attached to $T$. By the inductive hypothesis, we know that without $w$, $(T, d_T)$ is consistent on with $d$ so we only need to show that it is consistent with the addition of $w$.

Now let $x, y, z$ be the universal tree used to sort $w$ in the penultimate recursive step. Let $r$ be the Steiner node. Then by Lemma 2, we know that $d_T(w, x) = d(w, x)$, $d_T(w, y) = d(w, y)$, and $d_T(w, z) = d(w, z)$.

Now without loss of generality assume that $w$ was sorted in a zone for $x$. That is, $w \in Zone_i(x)$ for $i = 1, 2$.

**Case 1:** If $w \in Zone_1(x)$. Then from Lemma 1, we know that for all $\hat{w} \notin Zone_1(x)$, we have that $x \in g(w, \hat{w})$. Thus, having $d_T(x, w) = d(x, w)$ and $d_T(x, \hat{w}) = d(x, \hat{w})$ is sufficient to show consistency.

Now, since $w$ was placed last there is at most one other point $\tilde{w}$ in $Zone_1(x)$, and $d_T(w, \tilde{w}) = d(w, \tilde{w})$ due to Lemma 1.

**Case 2:** If $w \in Zone_2(x)$. Then from Lemma 2, we know that for all $\hat{w} \notin Zone_i(x)$, for $i = 1, 2$ we have that $r \in g(w, \hat{w})$. Thus, having $d_T(r, w) = d(r, w)$ and $d_T(r, \hat{w}) = d(r, \hat{w})$ is sufficient to show consistency.

Suppose $\hat{w} in Zone_1(x)$. Then from Lemma 1, we have that $x \in g(w, \hat{w})$. Thus, having $d_T(x, w) = d(x, w)$ and $d_T(x, \hat{w}) = d(x, \hat{w})$ is sufficient to show consistency.

Finally, since $w$ was the last node placed there are no other nodes in $Zone_2(x)$.

Thus, we have the the tree returned by TREEREP is consistent with the input metric $d$.

Notice that whenever we add a Steiner node $r$ we fix the position of at least one data point node. We then look at $O(n)$ Gromov inner products. Thus, we have a worst case running time of $O(n^2)$.

Additionally, the part where we place nodes into their respective zones can be done in parallel. Thus, if we have $K$ threads then the running time is $O\left(\frac{n^2}{K}\right)$ for the worst running times.

The final part of the theorem is that we return the tree with the smallest possible nodes. Whenever we look at any triangle formed by three points $x, y, z$, we place a Steiner node $r$. Now, if none of the distances from $x, y, z$ to $r$ is 0, then this Steiner node must exist in all tree consistent with $d$. If one of these distances is 0, we contracted that edge and got rid of $r$. Thus, along with the local consistency argument above this shows that all Steiner nodes that we have placed are necessary (the local consistency argument implies that no two of the Steiner nodes placed could in fact be made into one node due to the nodes beings in different regions). Thus, we have the fewest possible nodes.

$\square$

## 7.2 Tree Approximation Proofs

**Proposition 1.** *Given a $\delta$-hyperbolic metric $d$, the universal tree $T$ on $x, y, z$ and a fourth point $w$, when sorting $w$ into its zone $zone_i(\pi x)$, TREEREP introduces an additive distortion of $\delta$ between $w$ and $\pi y, \pi z$*

*Proof.* Without loss of generality assume that $\pi$ is the identity. In this case, we know that $d_T(w, r) = (y, z)_w$, and that $d_T(y, r) = (x, z)_y$. Thus, we have the following:

$$\begin{aligned}
|d_T(w,y) - d(w,y)| &= |d_T(w,r) + d_T(r,y) - d(w,y)| \\
&= |(y,z)_w + (x,z)_y - d(w,y)| \\
&= \frac{1}{2}|d(w,z) + d(y,x) \\
&\quad - d(w,y) - d(x,y)| \\
&= |(x,y)_w - (x,z)_w| \\
&\leq \delta
\end{aligned}$$

$\square$

# 8 Geometry: Asymptotic Cones

**Definition 14.** *An ultrafilter $\mathcal{F}$ on $X$ is a subset of $\mathcal{P}(X)$ such that*

1. *If $A \in \mathcal{F}$ and $A \subset B$ then $B \in \mathcal{F}$*
2. *$A, B \in \mathcal{F}$ then $A \cap B \in \mathcal{F}$*
3. *For any $A \subset X$, exactly 1 of $A, X \setminus A$ is in $\mathcal{F}$*
4. *$\emptyset \notin \mathcal{F}$.*

One way to view $\mathcal{F}$ is as defining a probability measure on $X$. In particular, we will view the sets in $\mathcal{F}$ to be large and the sets not in $\mathcal{F}$ to be small. Hence, we can define a measure $\nu$ such that for all $A \in \mathcal{F}$ we have that $\nu(A) = 1$ and for all $A \notin \mathcal{F}$ we have that $\nu(A) = 0$.

In this way, we can see that $\nu$ is a finitely additive measure on $X$. One common method to define ultrafilters is to take a point $x \in X$ and let $\mathcal{F}$ be the set of all sets that contain $x$. In this case, the measure $\nu$ has a point mass at $x$ and zero mass elsewhere. Such filters are known an principal ultrafilters.

Given a measure $\nu$ on $\mathbb{N}$, we can use it to define limits and convergence in $X$. In particular, we have that a sequence $x_i$ converges to $x$, if for all $\epsilon > 0$ we have that

$$\nu\left(\{x_i : |x_i - x| < \epsilon\}\right) = 1$$

We will denote limits of this form as $\lim_\nu x_i = x$.

We will make use of ultrafilters to construct the asymptotic cone. We will do this via looking at a non-principal ultrafilter on $\mathbb{N}$. We consider non-principal ultrafilters as we want to get a view from infinity, and we do not want to be in the case when one particular index in $\mathbb{N}$ has the entire mass. Hence we restrict ourselves to non-principal ultrafilters. One nice characterization of non-principal ultrafilters is that they are exactly the ultrafilters that have no finite sets.

Now that we have mathematical framework in which we can take limits, let us define our asymptotic cone. Let $\omega$ be a non-principal ultrafilter on $\mathbb{N}$. Let $\{b_i\}_{i \in \mathbb{N}}$ be a sequence of base points and let $\{\lambda_i\}_{i \in \mathbb{N}}$ be a sequence of scaling factors that go to infinity. Let $d$ be the metric on our space $X$. Then let

$$X_{\omega, b_i, \lambda_i} = \{\{y_i\} : y_i \in X \text{ and } d(b_i, y_i) \leq const_{\{y_i\}}\lambda_i\}$$

While this space looks huge we will define an equivalence relation and mod out by this relation to obtain better structure on this space. Given two points $y = \{y_i\}, z = \{z_i\} \in X_{\omega, b_i, \lambda_i}$ we say that $y \sim z$ if

$$\lim_\omega \frac{d(y_i, z_i)}{\lambda_i} = 0$$

We can now define our asymptotic cone $Con_\omega(X) = X(\omega, b_i, \lambda_i)/\sim$. We can also define a metric on this space as follows, given $y = \{y_i\}, z = \{z_i\} \in Con_\omega(X)$

$$d_\omega(y, z) := \lim_\omega \frac{d(y_i, z_i)}{\lambda_i}$$

Let us look at a few examples to get a handle on what $Con_\omega(X)$ looks like.

1. Example 1: Let us first consider $X = \mathbb{R}^n$. We know that $\mathbb{R}^n$ is scale invariant. This results in $Con_\omega(\mathbb{R}^n)$ being equivalent to $\mathbb{R}^n$. In fact, if we assume that $b_i \equiv 0$, then the map $x \mapsto \{\lambda_i x\}$ is an isometry from $\mathbb{R}^n$ to $Con_\omega(\mathbb{R}^n)$

2. Example 2: Suppose $X$ is a bounded metric space. In this case $Con_\omega(X)$ is a single point.

**Definition 15.** *A metric space $(X, d_x)$ can be isometrically embedded into a metric space $(Y, d_y)$ if there exists a map $f : X \to Y$ such that for all $x_1, x_2 \in X$ we have that*

$$d_x(x_1, x_2) = d_y(f(x_1), f(x_2))$$

*Such a map $f$ is known as an isometry.*

**Definition 16.** *A metric space $(X, d)$ is homogenous if for all $x, y \in X$ there exists an isometry $f : X \to X$ such that $f(x) = y$.*

**Definition 17.** *Given a $\mathbb{R}$-tree $T$, the valency of a point $x \in T$ in an $\mathbb{R}$-tree is the number of connected components in $T \setminus \{x\}$. Let the valence of a the tree, denoted $val(T)$, be the maximum valence of any point in $T$.*

**Definition 18.** *A $\mathbb{R}$-tree $T$ is a $\mu$-universal if every $\mathbb{R}$-tree $\hat{T}$ with $val(\hat{T}) \leq \mu$ can be isometrically embedded into $T$.*

Here we can see that we can embed any finite tree into a $2^{\aleph_0}$-universal tree $T$. Hence, if could isometrically embed $T$ into $Con(\mathbb{H}^n)$ then we can embed any tree into $Con(\mathbb{H}^n)$. This and more turns out to be true.

**Theorem 4.** *[15] Any $2^{\aleph_0}$-universal $\mathbb{R}$-tree can be isometrically embedded into the asymptotic cone for any complete simply connected manifold of negative curvature.*

# 9 Geometry: Geodetic Tree

In general, it is rare to be able isometrically embed one space into another. Hence, we have the following weaker definition.

**Definition 19.** *We say that we can quasi isometrically embed a metric space $(X, d_x)$ into a metric space $(Y, d_y)$ if there exists a map $f : X \to Y$ and real numbers $c, \lambda \in \mathbb{R}$ such that $\lambda \geq 1$, $c > 0$ and for all $x_1, x_2 \in X$ we have that*

$$\frac{1}{\lambda} d_x(x_1, x_2) - c \leq d_y(f(x_1), f(x_2)) \leq \lambda d_x(x_1, x_2) + c$$

Such isometries are called $(\lambda, c)$-quasi-isometries.

It is has been shown that any $\delta$-hyperbolic metric space $(X, d)$ with bounded growth admits a quasi-isometric embedding into $\mathbb{H}^k$ [6].

**Definition 20.** *We say that a ray $R$ is quasi geodetic if instead of being an isometric image of $[0, \infty)$, we have that $R$ is an quasi-isometric image of $[0, \infty)$.*

**Definition 21.** *A ray is eventually (quasi) geodetic if it has a subray that is (quasi) geodetic.*

**Theorem 5.** *[22] For all $\lambda \geq 1, c \geq 0$ there is a constant $\kappa = \kappa(\delta, \lambda, c)$, such that for every two points $x, y \in \mathbb{H}^k$, every $(\lambda, c)$-quasi-geodesic between them lies in a $\kappa$-neighborhood around every geodesic between $x$ and $y$ and vice versa.*

**Definition 22.** *Two geodetic rays $\pi_1, \pi_2$ are equivalent if for any sequence $(x_n)$ of points on $\pi_1$, we have $\liminf_{n \to \infty} d(x_n, \pi_2) \leq M$ for an $M < \infty$*

**Definition 23.** *The boundary $\partial \mathbb{H}^k$ of $\mathbb{H}^k$ is the equivalence class of all geodesic rays.*

**Theorem 6.** *[22] There is an $\mathbb{R}$-tree $T \subset \mathbb{H}^k$ such that the canonical map $\gamma$ from $\partial T$ to $\partial X$ exists and has the following properties.*

1. *It is surjective;*

2. *there is a constant $M < \infty$ depending only on $k$ such that $\gamma^{-1}(\eta)$ has at most $M$ elements for each $\eta \in \partial \mathbb{H}^k$.*

**Theorem 7.** *[22] Let $T$ be the $\mathbb{R}$-tree in Theorem 6 with root $r$. There exist constants $\lambda \geq 1$, $c \geq 0$ such that every ray in $T$ starting at the root is a $(\lambda, c)$-quasi-geodetic ray in $\mathbb{H}^k$.*

The above two theorems tell us that given any geodesic ray $R$ in $\mathbb{H}^k$ there is exists a ray in $T$ that is equivalent to $R$ (via $\sim$ in Definition 22). Furthermore this ray in $T$ is $(\lambda, c)$-quasi-geodetic ray in $\mathbb{H}^k$. Thus, due to Theorem 5 any configuration of points at scale in $\mathbb{H}^k$ can be approximated by a tree such that the larger the scale, better the approximation.

## 10 TREEREP Best

So far all numbers for the TREEREP algorithm that we have reported are averages. But due to the speed of the algorithm, we can actually run the experiment multiple times and pick the tree with the best metric.

Table 5: TREEREP Best Numbers

| | No Opt | | Heuristic Opt | | Full Opt | |
|---|---|---|---|---|---|---|
| Graph | MAP | Distortion | MAP | Distortion | MAP | Distortion |
| Celegan | 0.508 | 0.173 | 0.539 | 0138 | 0.547 | 0.119 |
| Diseasome | 0.912 | 0.134 | 0.911 | 0.106 | 0.890 | 0.092 |
| CS PhD | 0.987 | 0.134 | 0.984 | 0.119 | 0.968 | 0.121 |
| Yeast | 0.841 | 0.171 | 0.833 | 0.150 | 0.808 | 0.135 |
| Grid-worm | 0.727 | 0.154 | 0.728 | 0.125 | - | - |
| GRQC | 0.699 | 0.175 | 0.694 | 0.152 | - | - |

## 11 Improving Distortion

We have seen that in the case of unweighted graphs TREEREP produces better MAP than PM, LM, and PT. However, PT tends to have better average distortion. Hence, we want to be able to improve the distortion. Once we have learned the tree structure we can set up an optimization problem to learn the edge weights on the tree to improve the distortion. Specifically, since the metric comes from the tree, for any pair of data points, there is exactly one path connecting the two data points. Thus, regardless of the edges weights, this path is the shortest path between the data points. Thus, we can set up an optimization problem of the following form:

$$\arg \min_w \|AW - D\|_2.$$

Here $W$ is a vector containing the edge weights, $D$ is a vector containing the original metric, and $A$ is a matrix that encodes all of the paths. This optimization problem however, is unfeasible as $n$ gets longer. So instead we sample some rows of $A$ and solve a heuristic problem. As can be seen from Table 6, we are still faster than NJ but now have improved our distortion without sacrificing MAP.

Table 6: MAP and average distortion for the TREEREP and MST after doing the heuristic optimization. The time taken for both optimizations is the same.

| Graph | Time | Distortion | MAP | Distortion | MAP |
|---|---|---|---|---|---|
| | | TREEREP | | MST | |
| Celegans | 0.69 | 0.157 | 0.504 | 0.195 | 0.357 |
| Diseasome | 1.56 | 0.121 | 0.891 | 0.111 | 0.774 |
| CS Phd | 1.2 | 0.152 | 0.971 | 0.170 | 0.989 |
| Yeast | 4.2 | 0.163 | 0.813 | 0.171 | 0.862 |
| Grid Worm | 32 | 0.164 | 0.707 | 0.151 | 0.768 |
| GRQC | 68 | 0.157 | 0.676 | 0.159 | 0.669 |

## 12  Experiment and Practical Details

### 12.1  MAP and Average Distortion

**Definition 24.** *Given two metrics $d_1, d_2$ on a finite set $X = x_1, \ldots, x_n$ the average distortion is:*

$$\frac{1}{\binom{n}{2}} \sum_{i=1}^{n} \sum_{j<i} \frac{|d_1(x_i, x_j) - d_2(x_i, x_j)|}{d_2(x_i, x_j)}$$

*Smaller average distortion implies greater similarity between $d_1$ and $d_2$.*

In many cases, the metric learned by the various algorithms will be a scalar multiple of the actual metric, so we will solve for the scale $\alpha := \arg\min_c \|D - c\hat{D}\|_F$, before calculating the average distortion.[7]

**Definition 25.** *Let $d$ be a metric on the nodes of a graph $G = (V, E)$. For $v \in V$, let $N(v) = \{u_1, \ldots, u_{deg(v)}\}$ be the neighborhood of $v$. Then let $B_{v,u_i} = \{u \in V \setminus \{u\} : d(u, v) \leq d(v, u_i)\}$. Then the mean average precision (MAP) is defined to be*

$$\frac{1}{n} \sum_{v \in V} \frac{1}{deg(v)} \sum_{i=1}^{|N(v)|} \frac{|N(v) \cap B_{v,u_i}|}{|B_{v,u_i}|}$$

*Closer MAP is to 1, the closer $d$ is to approximating $d_G$.*

### 12.2  TreeRep

There are a few practical details that must be discussed in relation to the TREEREP algorithm.

1. Pre-allocate the matrix for the weights of edges of the tree as a dense matrix. Doing this greatly speeds up computations. Note the proof of Lemma 2, show that we need at most $n$ Steiner nodes. Thus, the tree has about $2n$ nodes. Since the input to the algorithm is a dense $n \times n$ matrix, we already need $O(n^2)$ memory. Thus, having a dense $2n \times 2n$ matrix is still linear memory usage in the size of the input.
2. When doing zone 2 recursions pick the node closest to $r$ as the new $z$ as suggested by Proposition 1.
3. The placement of nodes into their respective zones can be done in parallel. For all of the experiments in the paper, we used 8 threads to do the placement for all of the experiments, except that we used 1 thread for the random points from $\mathbb{H}^k$ experiment and for CBMC experiment.
4. All of the numbers reported are averages over 20 iterations. We could have also picked the best over 20 iterations as our algorithm is fast enough for this to be viable.
5. When checking for equality, instead of checking for exact equality, we checked whether two numbers are within 0.1 of each other.
6. It is possible for some of the edge weights to be set to a negative number. In this case, after the algorithm terminated we set those edge weights to 0.

### 12.3  Bartal

We sample 200 trees from the distribution and compute the metric assuming that we are embedding into the distribution restricted to these 200 trees.

### 12.4  Neighbor Join

The following implementation of NJ was used: http://crsl4.github.io/PhyloNetworks.jl/latest/. We set the options so as to not have any negative edge weights.

## 12.5 MST

Prim's algorithm for calculating MST was used. We used the implementation at https://github.com/JuliaGraphs/LightGraphs.jl

## 12.6 LS

Low stretch spanning trees are calculated using Laplacian package in Julia. This code is based an adaptation of [2] by the authors of [16].

## 12.7 LevelTree and ConstructTree

To the best of the authors knowledge there does not exist a publicly available implementations of these algorithms. Both of these algorithms were implemented by the authors.

Note that LevelTree claims to be a $O(n)$ algorithm, but this only true, once we have calculated the sphere $S_n$ needed for the algorithm. However, it takes $O(n^2)$ time to calculate the spheres $S_n$ (equivalent to solving single source all destination shortest path problem).

## 12.8 PM and LM

The following options were used. The number of epochs was to set to be higher than default. Everything else was left at default. One note about PM and LM is that their objective function is set up to optimize for MAP and not average distortion.

1. -lr 0.3
2. -epochs 1000
3. -burnin 20
4. -negs 50
5. -fresh
6. -sparse
7. -train_threads 2
8. -ndproc 4
9. -batchsize 10

For PM we used `-manifold poincare`, for LM we used `-manifold lorentz`. The code is taken from https://github.com/facebookresearch/poincare-embeddings

## 12.9 PT

The following options were used. We used the `-learn-scale` option as based on the discussion in the appendix of Sala et al. [36] learning the scale results in better quality metrics. Additionally, we add a burnin phase to the optimization. Finally, based on the discussion in [36], the objective function for PT has a lot of shallow local minimas. Thus, we added momentum and used Adagrad for the optimization to try and avoid these local minimums.

1. --learn-scale
2. --burn-in 100
3. --momentum 0.9
4. --use-adagrad
5. --l 5.0
6. --epochs 1000
7. --batch-size 256
8. --subsample 64

The code is taken from https://github.com/HazyResearch/hyperbolics

## 12.10 Hardware

All experiments were run on Google cloud instances. For PM, LM and PT we created a fresh instance for each algorithm. Each instance for an algorithm only had the bare minimum installed to run those

algorithms. We used `n1-highmem-8` instances. The specification of each of the instances are as follows:

1. 8 cores each with 6.5 GB of ram.
2. Ubuntu-1604-xenial-v20190913 operting system.
3. 100 standard persistent disk.

For TreeRep, NJ, CT, LT and MST, we ran all code via a Jupyter notebook interface running Julia 1.1.0. All experiments (except for the experiments with Enron and Wordnet), we done on instances with the same specification as above.

For Enron and Wordnet, we need more memory to store the distance matrices. Thus, used an since with the following specifications.

1. 24 cores each with 6.5 GB of ram.
2. Ubuntu-1604-xenial-v20190913 operting system.
3. 100 standard persistent disk.

## 12.11 Synthetic $0$-hyperbolic metrics

To produce random synthetic 0-hyperbolic metrics, we do the following. First, we take a complete binary tree of depth $i$. We then compute its double tree. Then for each node in this tree we sample a number $C$ from 2 to 10 and replace the node with a clique of size $C$. We then pick a random node in the tree and compute the breadth first search tree from that node. We then assign edge uniformly randomly, sampled from $[0, 1]$.

## 12.12 Synthetic Data Sets

Here we sampled coordinates from the standard normal $\mathcal{N}(0, 1)$. The final coordinate $x_0$ is set so that the point lies on the hyperboloid manifold. In the presence of a scale we just multiplied each coordinate by that scale before calculating $x_0$. We ran TREEREP with 1 thread.

## 12.13 Phylogenetic and Single Cell Data

The immunological distances can be seen in Figure 5. The matrix is symmeterized by averaging across the diagonal. In this case, we ran TREEREP 10 times and picked the tree with the lowest average distortion.

The figures for the trees are produced using an adaptation of Sarkar's construction for Euclidean space. The code from PT also produces a picture. This picture can be seen in Figure 4. As we can see, this figure is similar to the one in the main text.

For the Zeisel data we did the same pre-processing as done in Dumitrascu et al. [14]. For PM and MST, we use 10 nearest neighbor graph. For LS we used the complete graph.

For the CBMC data we did the same pre-processing as done in Dumitrascu et al. [14]. For MST and LS we used the complete graph.

## 12.14 Unweighted Graphs

Some of the graphs are disconnected. The largest connected component of each graph was used.

For $\delta$ calculation, we normalized the distances so that the maximum distance was 1 and then calculated $\delta$. For Celegans, Diseasome, and Phds,, this calculation is exact.

For Yeast, Grid-worm and GRQC, we fixed the base point to be $w = 1$ and then calculated $\delta$. It is known from theory that for any fixed base point the $\delta$ is at least half of the $\delta$ for the whole metric [7]. Thus, we get the inequality.

All experiments with a "-" were terminated after 4 hours.

## 12.15 Calculating $\alpha$

Can be calculated directly using

Figure 4: Figure for Sarich data produced by PT code

```
dog           0   32   48   51   50   48   98  148
bear         32    0   26   34   29   33   84  136
raccoon      48   26    0   42   44   44   92  152
weasel       51   34   42    0   44   38   86  142
seal         50   29   44   44    0   24   89  142
sea lion     48   33   44   38   24    0   90  142
cat          98   84   92   86   89   90    0  148
monkey      148  136  152  142  142  142  148    0
```

Figure 5: Immunological distances from [37]

$$\alpha = \frac{Tr(D' * D)}{\|D\|_F^2}$$

## 13 Tree Representation Pseudo-code

We can see examples of what happens when we set the new Steiner node for the two different kinds of recursion in Figure 6

Table 7: Table with the time taken in seconds, MAP, and average distortion for all of the algorithms when given metrics that come from unweighted graph. Darker cell colors indicates better numbers for MAP and average distortion. The number next to PT, PM, LM is the dimension of the space used to learn the embedding. The numbers for TREEREP (TR) are the average numbers over 20 trials. The table also shows some graph statistics such as $n$, the number of nodes, $m$, the number of edges, and $\delta$, the hyperbolcity of the metric.

| Graph | | TR | NJ | MST | LT | CT | LS | Bartal | PT 2 | PT 200 | PM 2 | LM 2 | LM 200 | PM 200 |
|---|---|---|---|---|---|---|---|---|---|---|---|---|---|---|
| | $n$ | | | | | | MAP | | | | | | | |
| Celegan | 452 | 0.473 | 0.713 | 0.337 | 0.272 | 0.447 | 0.313 | 0.436 | 0.098 | 0.857 | 0.479 | 0.466 | 0.646 | 0.662 |
| Dieseasome | 516 | 0.895 | 0.962 | 0.789 | 0.725 | 0.815 | 0.785 | 0.610 | 0.392 | 0.868 | 0.799 | 0.781 | 0.874 | 0.886 |
| CS Phd | 1025 | 0.979 | 0.993 | 0.991 | 0.964 | 0.807 | 0.991 | 0.190 | 0.190 | 0.556 | 0.537 | 0.537 | 0.593 | 0.593 |
| Yeast | 1458 | 0.815 | 0.892 | 0.871 | 0.742 | 0.859 | 0.873 | - | 0.235 | 0.658 | 0.522 | 0.513 | 0.641 | 0.643 |
| Grid-worm | 3337 | 0.707 | 0.800 | 0.768 | 0.657 | - | 0.766 | - | - | - | 0.334 | 0.306 | 0.558 | 0.553 |
| GRQC | 4158 | 0.685 | 0.862 | 0.686 | 0.480 | - | 0.684 | - | - | - | 0.589 | 0.603 | 0.783 | 0.784 |
| Enron | 33695 | 0.570 | - | 0.524 | - | - | 0.523 | - | - | - | - | - | - | - |
| Wordnet | 74374 | 0.984 | - | 0.989 | - | - | 0.989 | - | - | - | - | - | - | - |
| | $m$ | | | | | | Average Distortion | | | | | | | |
| Celegan | 2024 | 0.197 | 0.124 | 0.255 | 0.166 | 0.325 | 0.353 | 0.220 | 0.236 | 0.096 | 0.236 | 0.249 | 0.224 | 0.211 |
| Dieseasome | 1188 | 0.188 | 0.161 | 0.161 | 0.157 | 0.315 | 0.228 | 0.330 | 0.227 | 0.05 | 0.323 | 0.328 | 0.335 | 0.332 |
| CS Phd | 1043 | 0.204 | 0.134 | 0.298 | 0.161 | 0.282 | 0.291 | 0.326 | 0.295 | 0.105 | 0.374 | 0.378 | 0.378 | 0.380 |
| Yeast | 1948 | 0.205 | 0.149 | 0.243 | 0.243 | 0.282 | 0.243 | - | 0.230 | 0.089 | 0.246 | 0.248 | 0.234 | 0.234 |
| Grid-worm | 6421 | 0.188 | 0.135 | 0.171 | 0.202 | - | 0.234 | - | - | - | 0.196 | 0.203 | 0.192 | 0.193 |
| GRQC | 13422 | 0.192 | 0.200 | 0.275 | 0.267 | - | 0.206 | - | - | - | 0.212 | 0.198 | 0.193 | 0.193 |
| Enron | 180810 | 0.453 | - | 0.607 | - | - | 0.562 | - | - | - | - | - | - | - |
| Wordnet | 75834 | 0.131 | - | 0.336 | - | - | 0.071 | - | - | - | - | - | - | - |
| | $\delta$ | | | | | | Time in seconds | | | | | | | |
| Celegan | 0.21 | 0.014 | 0.28 | 0.0002 | 0.086 | 0.9 | 0.001 | 226 | 573 | 1156 | 712 | 523 | 1578 | 1927 |
| Dieseasome | 0.17 | 0.017 | 0.41 | 0.0003 | 0.39 | 15.76 | 0.001 | 313 | 678 | 1479 | 414 | 365 | 978 | 1112 |
| CS Phd | 0.23 | 0.037 | 2.94 | 0.0007 | 1.97 | 226 | 0.006 | 3559 | 1607 | 4145 | 467 | 324 | 768 | 1149 |
| Yeast | $\leq 0.32$ | 0.057 | 8.04 | 0.0008 | 8.21 | 957 | 0.001 | - | 9526 | 17876 | 972 | 619 | 1334 | 2269 |
| Grid-worm | $\leq 0.38$ | 0.731 | 163 | 0.001 | 191 | - | 0.007 | - | - | - | 2645 | 1973 | 4674 | 5593 |
| GRQC | $\leq 0.36$ | 0.42 | 311 | 0.0014 | 70.9 | - | 0.006 | - | - | - | 7524 | 7217 | 9767 | 1187 |
| Enron | - | - | 27 | - | 0.013 | - | - | 0.13 | - | - | - | - | - | - |
| Wordnet | - | - | 74 | - | 0.18 | - | - | 0.08 | - | - | - | - | - | - |

---

**Algorithm 3** Recursive parts of TreeRep.

---

1: **function** ZONE1_RECURSION($T, d_T, d, L, v$)
2:     **if** Length($L$) == 0 **then**
3:         **return** $T$
4:     **if** Length($L$) == 1 **then**
5:         Set $u$ = pop($L$) and add edge $(u, v)$ to $E$
6:         Set edge weight $d_T(u, v) = d(u, v)$
7:         **return** $T$
8:     Set $u$ =pop($L$), $z$ =pop($L$)
9:     **return** RECURSIVE_STEP($T, L, v, u, z, d, d_T$)
10:
11: **function** ZONE2_RECURSION($T, d_T, d, L, u, v$)
12:     **if** Length($L$) == 0 **then return** $T$
13:     Set $z$= the closest node to $v$.
14:     Delete edge $(u, v)$
15:     **return:** RECURSIVE_STEP($T, L, v, u, z, d, d_T$)

---

**Algorithm 4** Metric to tree structure algorithm.

---

 1: **function** TREE STRUCTURE(X, $d$)
 2:     $T = (V, E, d') = \emptyset$
 3:     Pick any three data points uniformly at random $x, y, z \in X$.
 4:     $T$ = RECURSIVE_STEP($T, X, x, y, z, d, d_T$,)
 5:     **return** $T$

 6:

 7:

 8: **function** RECURSIVE_STEP($T, X, x, y, z, d, d_T$,)
 9:     Let $Z1(r \to [], x \to [], y \to [], z \to []), Z2(x \to [], y \to [], z \to [])$    // Dictionaries of list
    for various zones
10:     Place an additional node $r$ in $V$ and add edges $xr, yr, zr$ to $E$
11:     Set the weights $d_T(x, r) = (y, z)_x$, $d_T(y, r) = (x, z)_y$, and $d_T(z, r) = (x, y)_z$ // If edge
    weight = 0, contract the edge.
12:     **for** all remaining data points $w \in X$ **do**
13:         $a = (x, y)_w$, $b = (y, z)_w$, $c = (z, x)_w$, $m = 0$, $m2 = 0$
14:         **if** $a == b == c$ **then**
15:             push($w, Z1[r]$)
16:             Set $d_T(w, r) = (x, y)_w$
17:         **else if** $a == maximum(a, b, c)$ **then**
18:             $\pi = (x \to z, y \to y, z \to x)$
19:             $m = b$, $m2 = c$
20:             Set $d_T(w, r) = a$
21:         **else if** $b == maximum(a, b, c)$ **then**
22:             $\pi = (x \to x, y \to y, z \to z)$
23:             $m = a$, $m2 = c$
24:             Set $d_T(w, r) = b$
25:         **else if** $c == maximum(a, b, c)$ **then**
26:             $\pi = (x \to y, y \to x, z \to z)$
27:             $m = a$, $m2 = b$
28:             Set $d_T(w, r) = c$
29:         **if** $d(w, \pi x) == m$ or $d(w, \pi x) == m2$ **then**
30:             push($w, Z1[\pi x]$)
31:         **else**
32:             push($w, Z2[\pi x]$)
        // recurse on each of the zones
33:     $T$ = ZONE1_RECURSION($T, d_T, d, Z1[r], r$)
34:     $T$ = ZONE1_RECURSION($T, d_T, d, Z1[x], x$)
35:     $T$ = ZONE1_RECURSION($T, d_T, d, Z1[y], y$)
36:     $T$ = ZONE1_RECURSION($T, d_T, d, Z1[y], z$)
37:     $T$ = ZONE2_RECURSION($T, d_T, d, Z2[x], x, r$)
38:     $T$ = ZONE2_RECURSION($T, d_T, d, Z2[y], y, r$)
39:     $T$ = ZONE2_RECURSION($T, d_T, d, Z2[z], z, r$)
        **return** $T$

---

Figure 6: Figure showing the placement of the Steiner node $R'$ for the Zone 1 and Zone 2 recursion. The nodes in orange are Steiner nodes and the nodes in green come from the data set $V$.

[Supplementary Material 2 · Appendix.pdf]

# 6 Metric First Discussion and Justification

Table 7 shows that for most of the data sets, learning a tree structure first and then embedding it into hyperbolic space, yields embeddings with better MAP and average distortion compared to methods that learn the embedding directly. One possible explanation for this phenomenon is that the optimization problems that seek the embeddings directly are not being solved optimally. That is, the algorithms get stuck at some local minimum. Another possibility is that there is a disconnect between the objective being optimized and the statistics calculated to judge the quality of the embeddings.

We propose that there are geometric facts about hyperbolic space that suggest embedding by first learning a tree is the correct approach. The tree-likeness of hyperbolic space has been studied from many different approaches. We present details from Hamann [22], Dyubina and Polterovich [15] and looks at the geometry of $\mathbb{H}^k$ at its two extremes; large scale and small scale. Since $\mathbb{H}^k$ is a manifold, we know that at small scales hyperbolic space looks like Euclidean space. Additionally, in the Poincare disk, the hyperbolic Riemannian metric is given by $\frac{4}{(1-x^2-y^2)^2}(dx^2 + dy^2)$ and is just a re-scaling of the Euclidean metric. Thus, at small scales, hyperbolic space is similar to Euclidean space.

Hence to take advantage of hyperbolic representations (i.e., why learn a hyperbolic representation instead of a Euclidean one), we want to embed data into $\mathbb{H}^k$ at scale. To study the large scale geometry of $\mathbb{H}^k$, we consider the asymptotic cone for hyperbolic space $Con(\mathbb{H}^k)$. In particular, we can think of the asymptotic cone as the "view of our space from infinitely far away". See the more detailed discussion in Appendix 8 for examples and complete definitions. The following connects $Con(\mathbb{H}^k)$ to $\mathbb{R}$-tree spaces.

**Theorem 2.** *[42] $Con(\mathbb{H}^k)$ is a complete $\mathbb{R}$-tree.*

Thus, we see that the large scale structure of hyperbolic space is a tree, indicating a strong connection between learning trees and learning hyperbolic embeddings. Furthermore, it can be shown that $Con(\mathbb{H}^k)$ is a $2^{\aleph_0}$-universal tree. That is, any tree with finitely many nodes can be embedded into $Con(\mathbb{H}^k)$ exactly. However, these are still embeddings into $Con(\mathbb{H}^k)$. We would like to study embeddings into $\mathbb{H}^k$.

**Definition 11.** *A metric space $(T, d_T)$ admits an isometric embedding at infinity into the space $(X, d_X)$ if there exists a sequence of positive scaling factors $\lambda_i \to \infty$ such that for every point $t \in T$, there exists an infinite sequence $\{x_t^i\}, i = 1, 2, \ldots$ of points in $X$ such that for all $t_1, t_2 \in T$ $\lim_{i \to \infty} d_X(x_{t_1}^i, x_{t_2}^i)/\lambda_i = d_T(t_1, t_2)$*

**Theorem 3.** *[15] $Con(\mathbb{H}^k)$ can be isometrically embedded at infinity into $\mathbb{H}^k$.*

Thus, we can embed any tree into $\mathbb{H}^k$ with arbitrarily low distortion. A type of converse is also true.

**Definition 12.** *A (geodesic) ray $R$ is a (isometric) homeomorphic image of $[0, \infty)$, such that for any ball $B$ of finite diameter, $R$ lies outside $B$ eventually.*

Hamann [22] showed that we can construct a rooted $\mathbb{R}$-tree $T$ inside $\mathbb{H}^k$, such that every geodesic ray in $\mathbb{H}^k$ eventually converges to a ray of $T$. Thus, showing that any configuration of points at scale in $\mathbb{H}^k$ can be approximated by a tree. Additionally, larger the scale points can be better approximated by trees. More details can be found in Appendix 9. Thus, showing that learning a tree and then embedding this tree into $\mathbb{H}^k$ is equivalent to learning hyperbolic representations at scale.

This provides an explanation for why as the scale and dimension increased, TREEREP found a tree that better approximated the hyperbolic metric in Section 4. This also provides a justification for why learning a tree first, results in better hyperbolic representations.

# 7 Proofs

## 7.1 Tree Representation Proofs

**Lemma 1.** *Given a metric $d$ on three points $x, y, z$, there exists a (weighted) tree $(T, d_T)$ on four nodes $x, y, z, r$, such that $r$ is adjacent to $x, y, z$, the edge weights are given by $w(x, r) = (y, z)_x$, $w(y, r) = (x, z)_y$ and $w(z, r) = (x, y)_z$, and the metric $d_T$ on the tree agrees with $d$.*

*Proof.* The basic structure of this tree can be seen in Figure 1(d). To prove that the metrics agree we such need to see the following calculation.

$$
\begin{aligned}
d_T(x,y) &= w(x,r) + w(r,y) \\
&= (y,z)_x + (x,z)_y \\
&= \frac{1}{2}(d(x,y) + d(x,z) - d(y,z) \\
&\quad + d(x,y) + d(y,z) - d(x,z)) \\
&= d(x,y)
\end{aligned}
$$

Here $d_T$ is the metric on the tree $T$. $\square$

One important fact that we need is that if $(X,d)$ is a metric graph, then for any three distinct points $x, y, z \in X$, the geodesics connecting them intersect at a unique point. As seen in Lemma 1, we refer to this point a Steiner point $r$. It is now important to note that even though $r$ may not be a point in the data set we are given, but $r \in X$ [7]. Thus, in the following lemmas, whenever we find a Steiner point, we will assume that the metric $d$ is defined on $r$.

**Lemma 4.** *If $d$ is a tree metric and $x, y, w$ are three points then*

1. $(x,y)_w = 0$ if and only if $w \in g(x,y)$
2. $(x,y)_w = d(x,w)$ if and only if $(w,y)_x = 0$.
3. $(x,y)_w = d(y,w)$ if and only if $(w,x)_y = 0$.

Here $g(x,y)$ is the unique path connecting $x$ and $y$.

*Proof.* For 1. we see that

$$
\begin{aligned}
0 = (x,y)_w &= \frac{1}{2}(d(w,x) + d(w,y) - d(x,y)) \\
&\Rightarrow d(x,y) = d(w,x) + d(w,y)
\end{aligned}
$$

Thus we have that $w \in g(x,y)$.

For 2. we see that

$$
\begin{aligned}
(x,y)_w = d(x,w) &\Rightarrow d(w,x) + d(w,y) - d(x,y) \\
&= 2d(w,x) \\
&\Rightarrow d(w,x) + d(x,y) - d(w,y) = 0 \\
&\Rightarrow 2(w,y)_x = 0
\end{aligned}
$$

The proof for 3 is similar to that of 2.

$\square$

**Lemma 2.** *Let $(X,d)$ be a tree space. Let $w, x, y, z$ be four points in $X$ and let $(T, d_T)$ be the universal tree on $x, y, z$ with node $r$ as the Steiner node. Then we can extend $(T, d_T)$ to $(\hat{T}, d_{\hat{T}})$ to include $w$ such that $d_{\hat{T}} = d$.*

*Proof.* We note that there are four different possible cases for the configuration of $x, y, z, w$ depending on the relationship amongst the Gromov products. Each case determines a different placement of $r$, as follows:

1. If $(x,y)_w = (x,z)_w = (y,z)_w = 0$, then replace $r$ with $w$ to obtain $\hat{T}$.
2. If $(x,y)_w = (x,z)_w = (y,z)_w = c > 0$, then connect $w$ to $r$ via an edge of weight $c$ to obtain $\hat{T}$.
3. If there exists a permutation $\pi : \{x,y,z\} \to \{x,y,z\}$ such that,

$$
(\pi x, \pi y)_w = (\pi x, \pi z)_w = c < (\pi y, \pi z)_w
$$

and $d(\pi x, w) = (\pi x, \pi y)_w$, then connect $w$ to $\pi x$ via an edge of weight $c$ to obtain $\hat{T}$.

4. If there exists a permutation $\pi : \{x, y, z\} \to \{x, y, z\}$ such that,

$$(\pi x, \pi y)_w = (\pi x, \pi z)_w = c < (\pi y, \pi z)_w$$

and $d(\pi x, w) > (\pi x, \pi y)_w$, then add a Steiner point $\hat{r}$ on the edge $x, r$ with $d(\pi x, \hat{r}) = d(\pi x, w) - c$ and connect $w$ to $\hat{r}$ via an edge of weight $c$ to obtain $\hat{T}$.

To prove that these extensions of $T$ are consistent, first let us prove that there are exactly four cases. To do that, first note that since we have a 0-hyperbolic metric, at least two of the three Gromov products must be equal. Using the triangle inequality, we can see that for any three points $a, b, c$ the following holds

$$0 \le (a, b)_c \le d(a, c).$$

That is, either we are in the first two cases and three of products are equal, or we have that two of the products are equal. In the case that two of the products are equal, the permutation $\pi$ tells us which of the two are equal and we further subdivide into the case whether $d(\pi x, w) = (\pi x, \pi y)_w$ or $d(\pi x, w) > (\pi x, \pi y)_w$ as we cannot have $d(\pi x, w) < (\pi x, \pi y)_w$.

Therefore, there are at most four possible configuration cases and it remains to show that the new tree $d_{\hat{T}}$ is consistent with $d$ on the four points. In each case, we present the high level intuition for why these modification result in a consistent tree. The low level details about the metric numbers can easily be checked.

**Case 1:** If $(x, y)_w = (x, z)_w = (y, z)_w = 0$, then we replaced $r$ with $w$ in $\hat{T}$. In this case, using Lemma 4, we see that $w$ must lie on all tree geodesics $g(x, y), g(x, z), g(y, z)$. Since the metric comes from a tree, these three geodesics can only intersect at one point $r$. Thus, we must replace $r$ with $w$.

To see that the metric is consistent, we need to verify that $d(w, x) = d_{\hat{T}}(r, x)$. To see we have the following:

$$\begin{aligned} d_{\hat{T}}(r, x) &= (y, z)_x \\ &= (y, z)_x + (x, y)_w + (x, z)_w - (y, z)_w \\ &= d(w, x) \end{aligned}$$

**Case 2:** If

$$(x, y)_w = (x, z)_w = (y, z)_w = c > 0,$$

then we can see that $(x, w)_r = (y, w)_r = (z, w)_r = 0$. In this case, $r$ lies on geodesics $g(x, y)$, $g(x, z)$, $g(x, w)$, $g(y, w)$, $g(y, z)$, $g(z, w)$. Thus, we must have a star shaped graph with $r$ in the center.

To see that the metric is consistent we just need to verify that $d(w, x) = d_{\hat{T}}(w, x)$. To see that we have the following calculation.

$$\begin{aligned} d_{\hat{T}}(w, x) &= d_{\hat{T}}(w, r) + d_{\hat{T}}(x, r) \\ &= (x, y)_w + (y, z)_x \\ &= (x, y)_w + (x, z)_w - (y, z)_w + (y, z)_x \\ &= d(w, x) \end{aligned}$$

**Case 3:** In this case suppose condition 4 is true. Without loss of generality assume that $\pi$ is the identity map. In each case, we have a tree that looks like a tree in Figure 1. In this case, we can do the calculations and see that $(w, y)_r = (w, z)_r = 0$. That is, the geodesics $g(w, y), g(w, z), g(y, z), g(x, y), g(x, z)$ all intersect at the same point. Thus, again telling us our tree structure.

To check that the metric is consistent, we need to verify that $d(w, y) = d_{\hat{T}}(w, y) = d_{\hat{T}}(w, r) + d_{\hat{T}}(r, y)$. Before we can do that, let us first verify that

$$d_{\hat{T}}(w, r) = (y, z)_w$$

To verify this we need to the following calculation

$$
\begin{aligned}
d_{\hat{T}}(w, r) &= d_{\hat{T}}(r, \hat{r}) + d_{\hat{T}}(\hat{r}, w) \\
&= c + d_T(x, r) - d_{\hat{T}}(x, \hat{r}) \\
&= c + (y, z)_x - (d(x, w) - c) \\
&= 2c + (y, z)_x - d(x, w) \\
&= (x, y)_w + (x, z)_w + (y, z)_x - d(w, x) \\
&= (y, z)_w
\end{aligned}
$$

We then can see that

$$
\begin{aligned}
d_{\hat{T}}(w, y) &= d_{\hat{T}}(w, r) + d_{\hat{T}}(r, y) \\
&= (y, z)_w + (x, z)_y \\
&= (y, z)_w + (x, y)_w - (x, z)_w + (x, z)_y \\
&= d(w, y)
\end{aligned}
$$

Note $d_{\hat{T}}(w, r) = (y, z)_w$ and the consistency of the metric implies that $d(w, r) = (y, z)_w$. Finally, we can see $(w, y)_r = 0$ as follows.

$$
\begin{aligned}
2(w, y)_r &= d(w, r) + d(r, y) - d(w, y) \\
&= (z, y)_w + (x, z)_y - d(w, y) \\
&= \frac{1}{2}(d(w, z) - d(w, y) + d(x, y) - d(x, z)) \\
&= (x, z)_w - (x, y)_w \\
&= 0
\end{aligned}
$$

Note that this also implies that $(w, x)_r > 0$.

**Case 4:** In this case, suppose condition 3 is true. Without loss of generality assume that $\pi$ is the identity map. Then in this case, we still have that $(w, y)_r = (w, z)_r = 0$, but in addition we have that $(w, y)_x = (w, z)_x = 0$. Thus, again telling us our tree structure.

In this case, to verify that the metric is consistent, we need to check that $d(w, y) = d_{\hat{T}}(w, y) = d_{\hat{T}}(w, x) + d_{\hat{T}}(x, y)$. To see this we have the following calculations.

$$
\begin{aligned}
d_{\hat{T}}(w, x) + d_{\hat{T}}(x, y) &= (x, y)_w + d(x, y) \\
&= 2(x, y)_w - (x, z)_w + d(x, y) \\
&= d(w, y) + (w, z)_x
\end{aligned}
$$

Thus, now it suffices to show that $(w, z)_x = 0$, which can be seen using the following calculations.

$$
\begin{aligned}
(x, z)_w = d(w, x) &\Rightarrow 0 = d(x, w) + d(x, z) - d(w, z) \\
&\Rightarrow (w, z)_x = 0
\end{aligned}
$$

This also implies that $(w, z)_r = 0$. $\qquad\square$

The proof of Lemma 2 shows that there are a number of ways to extend $T$ to include the new point $w$. To clarify our discussion of the extension of $T$, we introduce new terminology.

**Definition 13.** *Given a data set $V$ (consisting of data points, along with the distances amongst the points), a universal tree $T$ on $x, y, z \in V$ (with $r$ as the Steiner node), let us defining the following three zone types. The first type is associated only with the Steiner node $r$, while the other two types are defined for each of the original nodes $x, y, z$.*

    *1. $Zone_1(r)$ is all $w \in V$ such that condition 2 is true in Lemma 2.*

2. *For a given permutation $\pi$, $Zone_1(\pi x)$ is all $w \in V$ such that condition 3 is true in Lemma 2 with $\pi$.*
3. *For a given permutation $\pi$, $Zone_2(\pi x)$ is all $w \in V$, such that condition 4 is true in Lemma 2 with $\pi$.*

Note that there are seven zones total.

**Lemma 5.** *Let $(X, d)$ is a metric tree. Let $x, y \in X$ and let $r \in g(x, y)$ if and only if $X \setminus \{r\}$ has at least two disconnected components and $x, y$ are in distinct components.*

*Proof.* Suppose $r \in g(x, y)$. In metric trees, we know that there exist unique simple path between any two points. Therefore, if, after removing $r$, a path connecting $x, y$ remained (i.e., they are in the same component), then there are two simple paths connecting $x, y$ in $X$, which is not possible.

Suppose $x, y$ are in two separate components of $X \setminus \{r\}$, then because $X$ is path connected, the geodesic between $x$ and $y$ must pass through $r$. $\qquad\square$

**Lemma 3.** *Given $(X, d)$ a metric tree, and a universal tree $T$ on $x, y, z$, we have the following*

1. *If $w \in Zone_1(x)$, then for all $\hat{w} \notin Zone_1(x)$, we have that $x \in g(w, \hat{w})$.*
2. *If $w \in Zone_2(x)$, then for all $\hat{w} \notin Zone_i(x)$ for $i = 1, 2$, then we have that $r \in g(w, \hat{w})$.*

*Proof.* First let us prove statement 1. To do this, let us analyze the possible zones to which $\hat{w}$ belongs.

**Case 1:** Suppose $\hat{w} \in Zone_1(y)$ (similar for $\hat{w} \in Zone_1(z)$). Then we have that $d(\hat{w}, y) = (x, y)_{\hat{w}}$. This, implies that $(\hat{w}, x)_y = 0$. Thus, by Lemma 4, we have that $y \in g(\hat{w}, x)$. Similarly we have that $x \in g(w, y)$.

Now since $w \in Zone_1(x)$, we know that $g(x, w) \cap g(x, y) = \{x\}$. Similarly, know that $g(x, y) \cap g(y, \hat{w}) = \{y\}$. Then using Lemma 5, on removing $x$, we see that $w$ and $y$ are different connected components. Then since $x \notin g(\hat{w}, y)$, we see that $\hat{w}, y$ is in one connected component. Thus, $w, \hat{w}$ are in different components. Thus, $x \in g(w, \hat{w})$ by Lemma 5.

**Case 2:** Suppose $\hat{w} \in Zone_2(y)$ (similar for $\hat{w} \in Zone_2(z)$). Now let $r$ be the Steiner node of the universal tree on $x, y, z$. In this case we know from Lemma 2 that $r \in g(\hat{w}, x)$ and that $g(w, x) \cap g(x, r) = \{x\}$.

Now since $w \in Zone_1(x)$, we know that $g(x, w) \cap g(x, r) = \{x\}$. Similarly, know that $g(x, r) \cap g(r, \hat{w}) = \{r\}$. Then using Lemma 5, on removing $x$, we see that $w$ and $r$ are different connected components. Then since $x \notin g(\hat{w}, r)$, we see that $\hat{w}, r$ is in one connected component. Thus, $w, \hat{w}$ are in different components. Thus, $x \in g(w, \hat{w})$ by Lemma 5.

**Case 3:** $\hat{w} \in Zone_2(x)$. Let $r$ be the Steiner node for the universal tree on $x, y, z$. Now my Lemma 2, we know that $x \in g(w, r)$. Thus, again by removing $x$ and using Lemma 5, $r$ and $w$ are in different. We also have that by Lemma 2 $x \notin g(\hat{w}, r)$. Thus $r, \hat{w}$ are in the same connected component of $X \setminus \{x\}$. Thus, $w$ and $\hat{w}$ are in different connected components. Thus, by Lemma 5, $x \in g(w, \hat{w})$

Thus in all cases, we can see that $x \in g(w, \hat{w})$

Now let us prove statement 2. Without loss of generality assume that

$$\hat{w} \in Zone_i(y)$$

for $i = 1, 2$. Then from Lemma 2, we know that $r \notin g(w, x)$ and $r \notin g(\hat{w}, y)$, but $r \in g(x, y)$. Thus, using Lemma 5 on removing $r$, $x$ and $y$ and in different components and $w$ is in the same component as $x$ and $\hat{w}$ is in the same component as $y$. Thus, again using Lemma 5, we have that $r \in g(w, \hat{w})$.

$\qquad\square$

**Theorem 1.** *Given $(X, d)$, a $\delta$-hyperbolic metric space, and $n$ points $x_1, \dots, x_n \in X$, TREEREP returns a tree $(T, d_T)$. In the case that $\delta = 0$, $d_T = d$, and $T$ has the fewest possible nodes. TREEREP has worst case run time $O(n^2)$. Furthermore the algorithm is embarrassingly parallelizable.*

*Proof.* The proof of this theorem follows directly from our structural lemmas. More precisely, we show that for $\delta = 0$, TREEREP returns a consistent metric via induction on $n$, the number of data points.

**Base Case:** The case when $n \leq 3$ is covered by Lemma 1. And, the case when $n = 4$ is covered by Lemma 2.

**Inductive Hypothesis:** Assume that for all $k \leq n$, our data set of $k$ points is consistent with a 0-hyperbolic metric $d$, then TREEREP returns a tree $(T, d_T)$ that is consistent with $d$ on the $k$ points.

**Inductive Step:** Assume that $w$ is the last vertex attached to $T$. By the inductive hypothesis, we know that without $w$, $(T, d_T)$ is consistent on with $d$ so we only need to show that it is consistent with the addition of $w$.

Now let $x, y, z$ be the universal tree used to sort $w$ in the penultimate recursive step. Let $r$ be the Steiner node. Then by Lemma 2, we know that $d_T(w, x) = d(w, x)$, $d_T(w, y) = d(w, y)$, and $d_T(w, z) = d(w, z)$.

Now without loss of generality assume that $w$ was sorted in a zone for $x$. That is, $w \in Zone_i(x)$ for $i = 1, 2$.

**Case 1:** If $w \in Zone_1(x)$. Then from Lemma 1, we know that for all $\hat{w} \notin Zone_1(x)$, we have that $x \in g(w, \hat{w})$. Thus, having $d_T(x, w) = d(x, w)$ and $d_T(x, \hat{w}) = d(x, \hat{w})$ is sufficient to show consistency.

Now, since $w$ was placed last there is at most one other point $\tilde{w}$ in $Zone_1(x)$, and $d_T(w, \tilde{w}) = d(w, \tilde{w})$ due to Lemma 1.

**Case 2:** If $w \in Zone_2(x)$. Then from Lemma 2, we know that for all $\hat{w} \notin Zone_i(x)$, for $i = 1, 2$ we have that $r \in g(w, \hat{w})$. Thus, having $d_T(r, w) = d(r, w)$ and $d_T(r, \hat{w}) = d(r, \hat{w})$ is sufficient to show consistency.

Suppose $\hat{w} in Zone_1(x)$. Then from Lemma 1, we have that $x \in g(w, \hat{w})$. Thus, having $d_T(x, w) = d(x, w)$ and $d_T(x, \hat{w}) = d(x, \hat{w})$ is sufficient to show consistency.

Finally, since $w$ was the last node placed there are no other nodes in $Zone_2(x)$.

Thus, we have the the tree returned by TREEREP is consistent with the input metric $d$.

Notice that whenever we add a Steiner node $r$ we fix the position of at least one data point node. We then look at $O(n)$ Gromov inner products. Thus, we have a worst case running time of $O(n^2)$.

Additionally, the part where we place nodes into their respective zones can be done in parallel. Thus, if we have $K$ threads then the running time is $O\left(\frac{n^2}{K}\right)$ for the worst running times.

The final part of the theorem is that we return the tree with the smallest possible nodes. Whenever we look at any triangle formed by three points $x, y, z$, we place a Steiner node $r$. Now, if none of the distances from $x, y, z$ to $r$ is 0, then this Steiner node must exist in all tree consistent with $d$. If one of these distances is 0, we contracted that edge and got rid of $r$. Thus, along with the local consistency argument above this shows that all Steiner nodes that we have placed are necessary (the local consistency argument implies that no two of the Steiner nodes placed could in fact be made into one node due to the nodes beings in different regions). Thus, we have the fewest possible nodes.

$\square$

## 7.2 Tree Approximation Proofs

**Proposition 1.** *Given a $\delta$-hyperbolic metric $d$, the universal tree $T$ on $x, y, z$ and a fourth point $w$, when sorting $w$ into its zone $zone_i(\pi x)$, TREEREP introduces an additive distortion of $\delta$ between $w$ and $\pi y, \pi z$*

*Proof.* Without loss of generality assume that $\pi$ is the identity. In this case, we know that $d_T(w, r) = (y, z)_w$, and that $d_T(y, r) = (x, z)_y$. Thus, we have the following:

$$\begin{aligned}
|d_T(w,y) - d(w,y)| &= |d_T(w,r) + d_T(r,y) - d(w,y)| \\
&= |(y,z)_w + (x,z)_y - d(w,y)| \\
&= \frac{1}{2}|d(w,z) + d(y,x) \\
&\quad - d(w,y) - d(x,y)| \\
&= |(x,y)_w - (x,z)_w| \\
&\leq \delta
\end{aligned}$$

$\square$

## 8 Geometry: Asymptotic Cones

**Definition 14.** *An ultrafilter $\mathcal{F}$ on $X$ is a subset of $\mathcal{P}(X)$ such that*

1. *If $A \in \mathcal{F}$ and $A \subset B$ then $B \in \mathcal{F}$*
2. *$A, B \in \mathcal{F}$ then $A \cap B \in \mathcal{F}$*
3. *For any $A \subset X$, exactly 1 of $A, X \setminus A$ is in $\mathcal{F}$*
4. *$\emptyset \notin \mathcal{F}$.*

One way to view $\mathcal{F}$ is as defining a probability measure on $X$. In particular, we will view the sets in $\mathcal{F}$ to be large and the sets not in $\mathcal{F}$ to be small. Hence, we can define a measure $\nu$ such that for all $A \in \mathcal{F}$ we have that $\nu(A) = 1$ and for all $A \notin \mathcal{F}$ we have that $\nu(A) = 0$.

In this way, we can see that $\nu$ is a finitely additive measure on $X$. One common method to define ultrafilters is to take a point $x \in X$ and let $\mathcal{F}$ be the set of all sets that contain $x$. In this case, the measure $\nu$ has a point mass at $x$ and zero mass elsewhere. Such filters are known an principal ultrafilters.

Given a measure $\nu$ on $\mathbb{N}$, we can use it to define limits and convergence in $X$. In particular, we have that a sequence $x_i$ converges to $x$, if for all $\epsilon > 0$ we have that

$$\nu\left(\{x_i : |x_i - x| < \epsilon\}\right) = 1$$

We will denote limits of this form as $\lim_\nu x_i = x$.

We will make use of ultrafilters to construct the asymptotic cone. We will do this via looking at a non-principal ultrafilter on $\mathbb{N}$. We consider non-principal ultrafilters as we want to get a view from infinity, and we do not want to be in the case when one particular index in $\mathbb{N}$ has the entire mass. Hence we restrict ourselves to non-principal ultrafilters. One nice characterization of non-principal ultrafilters is that they are exactly the ultrafilters that have no finite sets.

Now that we have mathematical framework in which we can take limits, let us define our asymptotic cone. Let $\omega$ be a non-principal ultrafilter on $\mathbb{N}$. Let $\{b_i\}_{i\in\mathbb{N}}$ be a sequence of base points and let $\{\lambda_i\}_{i\in\mathbb{N}}$ be a sequence of scaling factors that go to infinity. Let $d$ be the metric on our space $X$. Then let

$$X_{\omega, b_i, \lambda_i} = \{\{y_i\} : y_i \in X \text{ and } d(b_i, y_i) \leq const_{\{y_i\}} \lambda_i\}$$

While this space looks huge we will define an equivalence relation and mod out by this relation to obtain better structure on this space. Given two points $y = \{y_i\}, z = \{z_i\} \in X_{\omega, b_i, \lambda_i}$ we say that $y \sim z$ if

$$\lim_\omega \frac{d(y_i, z_i)}{\lambda_i} = 0$$

We can now define our asymptotic cone $Con_\omega(X) = X(\omega, b_i, \lambda_i)/\sim$. We can also define a metric on this space as follows, given $y = \{y_i\}, z = \{z_i\} \in Con_\omega(X)$

$$d_\omega(y, z) := \lim_\omega \frac{d(y_i, z_i)}{\lambda_i}$$

Let us look at a few examples to get a handle on what $Con_\omega(X)$ looks like.

1. Example 1: Let us first consider $X = \mathbb{R}^n$. We know that $\mathbb{R}^n$ is scale invariant. This results in $Con_\omega(\mathbb{R}^n)$ being equivalent to $\mathbb{R}^n$. In fact, if we assume that $b_i \equiv 0$, then the map $x \mapsto \{\lambda_i x\}$ is an isometry from $\mathbb{R}^n$ to $Con_\omega(\mathbb{R}^n)$

2. Example 2: Suppose $X$ is a bounded metric space. In this case $Con_\omega(X)$ is a single point.

**Definition 15.** *A metric space $(X, d_x)$ can be isometrically embedded into a metric space $(Y, d_y)$ if there exists a map $f : X \to Y$ such that for all $x_1, x_2 \in X$ we have that*

$$d_x(x_1, x_2) = d_y(f(x_1), f(x_2))$$

*Such a map $f$ is known as an isometry.*

**Definition 16.** *A metric space $(X, d)$ is homogenous if for all $x, y \in X$ there exists an isometry $f : X \to X$ such that $f(x) = y$.*

**Definition 17.** *Given a $\mathbb{R}$-tree $T$, the valency of a point $x \in T$ in an $\mathbb{R}$-tree is the number of connected components in $T \setminus \{x\}$. Let the valence of a the tree, denoted $val(T)$, be the maximum valence of any point in $T$.*

**Definition 18.** *A $\mathbb{R}$-tree $T$ is a $\mu$-universal if every $\mathbb{R}$-tree $\hat{T}$ with $val(\hat{T}) \leq \mu$ can be isometrically embedded into $T$.*

Here we can see that we can embed any finite tree into a $2^{\aleph_0}$-universal tree $T$. Hence, if could isometrically embed $T$ into $Con(\mathbb{H}^n)$ then we can embed any tree into $Con(\mathbb{H}^n)$. This and more turns out to be true.

**Theorem 4.** *[15] Any $2^{\aleph_0}$-universal $\mathbb{R}$-tree can be isometrically embedded into the asymptotic cone for any complete simply connected manifold of negative curvature.*

# 9 Geometry: Geodetic Tree

In general, it is rare to be able isometrically embed one space into another. Hence, we have the following weaker definition.

**Definition 19.** *We say that we can quasi isometrically embed a metric space $(X, d_x)$ into a metric space $(Y, d_y)$ if there exists a map $f : X \to Y$ and real numbers $c, \lambda \in \mathbb{R}$ such that $\lambda \geq 1$, $c > 0$ and for all $x_1, x_2 \in X$ we have that*

$$\frac{1}{\lambda} d_x(x_1, x_2) - c \leq d_y(f(x_1), f(x_2)) \leq \lambda d_x(x_1, x_2) + c$$

Such isometries are called $(\lambda, c)$-quasi-isometries.

It is has been shown that any $\delta$-hyperbolic metric space $(X, d)$ with bounded growth admits a quasi-isometric embedding into $\mathbb{H}^k$ [6].

**Definition 20.** *We say that a ray $R$ is quasi geodetic if instead of being an isometric image of $[0, \infty)$, we have that $R$ is an quasi-isometric image of $[0, \infty)$.*

**Definition 21.** *A ray is eventually (quasi) geodetic if it has a subray that is (quasi) geodetic.*

**Theorem 5.** *[22] For all $\lambda \geq 1, c \geq 0$ there is a constant $\kappa = \kappa(\delta, \lambda, c)$, such that for every two points $x, y \in \mathbb{H}^k$, every $(\lambda, c)$-quasi-geodesic between them lies in a $\kappa$-neighborhood around every geodesic between $x$ and $y$ and vice versa.*

**Definition 22.** *Two geodetic rays $\pi_1, \pi_2$ are equivalent if for any sequence $(x_n)$ of points on $\pi_1$, we have $\liminf_{n \to \infty} d(x_n, \pi_2) \leq M$ for an $M < \infty$*

**Definition 23.** *The boundary $\partial\mathbb{H}^k$ of $\mathbb{H}^k$ is the equivalence class of all geodesic rays.*

**Theorem 6.** *[22] There is an $\mathbb{R}$-tree $T \subset \mathbb{H}^k$ such that the canonical map $\gamma$ from $\partial T$ to $\partial X$ exists and has the following properties.*

1. *It is surjective;*

2. *there is a constant $M < \infty$ depending only on $k$ such that $\gamma^{-1}(\eta)$ has at most $M$ elements for each $\eta \in \partial\mathbb{H}^k$.*

**Theorem 7.** *[22] Let $T$ be the $\mathbb{R}$-tree in Theorem 6 with root $r$. There exist constants $\lambda \geq 1$, $c \geq 0$ such that every ray in $T$ starting at the root is a $(\lambda, c)$-quasi-geodetic ray in $\mathbb{H}^k$.*

The above two theorems tell us that given any geodesic ray $R$ in $\mathbb{H}^k$ there is exists a ray in $T$ that is equivalent to $R$ (via $\sim$ in Definition 22). Furthermore this ray in $T$ is $(\lambda, c)$-quasi-geodetic ray in $\mathbb{H}^k$. Thus, due to Theorem 5 any configuration of points at scale in $\mathbb{H}^k$ can be approximated by a tree such that the larger the scale, better the approximation.

## 10 TREEREP Best

So far all numbers for the TREEREP algorithm that we have reported are averages. But due to the speed of the algorithm, we can actually run the experiment multiple times and pick the tree with the best metric.

Table 5: TREEREP Best Numbers

| Graph | No Opt | | Heuristic Opt | | Full Opt | |
|---|---|---|---|---|---|---|
| | MAP | Distortion | MAP | Distortion | MAP | Distortion |
| Celegan | 0.508 | 0.173 | 0.539 | 0138 | 0.547 | 0.119 |
| Diseasome | 0.912 | 0.134 | 0.911 | 0.106 | 0.890 | 0.092 |
| CS PhD | 0.987 | 0.134 | 0.984 | 0.119 | 0.968 | 0.121 |
| Yeast | 0.841 | 0.171 | 0.833 | 0.150 | 0.808 | 0.135 |
| Grid-worm | 0.727 | 0.154 | 0.728 | 0.125 | - | - |
| GRQC | 0.699 | 0.175 | 0.694 | 0.152 | - | - |

## 11 Improving Distortion

We have seen that in the case of unweighted graphs TREEREP produces better MAP than PM, LM, and PT. However, PT tends to have better average distortion. Hence, we want to be able to improve the distortion. Once we have learned the tree structure we can set up an optimization problem to learn the edge weights on the tree to improve the distortion. Specifically, since the metric comes from the tree, for any pair of data points, there is exactly one path connecting the two data points. Thus, regardless of the edges weights, this path is the shortest path between the data points. Thus, we can set up an optimization problem of the following form:

$$\arg\min_w \|AW - D\|_2.$$

Here $W$ is a vector containing the edge weights, $D$ is a vector containing the original metric, and $A$ is a matrix that encodes all of the paths. This optimization problem however, is unfeasible as $n$ gets longer. So instead we sample some rows of $A$ and solve a heuristic problem. As can be seen from Table 6, we are still faster than NJ but now have improved our distortion without sacrificing MAP.

Table 6: MAP and average distortion for the TREEREP and MST after doing the heuristic optimization. The time taken for both optimizations is the same.

| Graph | Time | Distortion | MAP | Distortion | MAP |
|---|---|---|---|---|---|
| | | TREEREP | | MST | |
| Celegans | 0.69 | 0.157 | 0.504 | 0.195 | 0.357 |
| Diseasome | 1.56 | 0.121 | 0.891 | 0.111 | 0.774 |
| CS Phd | 1.2 | 0.152 | 0.971 | 0.170 | 0.989 |
| Yeast | 4.2 | 0.163 | 0.813 | 0.171 | 0.862 |
| Grid Worm | 32 | 0.164 | 0.707 | 0.151 | 0.768 |
| GRQC | 68 | 0.157 | 0.676 | 0.159 | 0.669 |

## 12 Experiment and Practical Details

### 12.1 MAP and Average Distortion

**Definition 24.** *Given two metrics $d_1, d_2$ on a finite set $X = x_1, \ldots, x_n$ the average distortion is:*

$$\frac{1}{\binom{n}{2}} \sum_{i=1}^{n} \sum_{j<i} \frac{|d_1(x_i, x_j) - d_2(x_i, x_j)|}{d_2(x_i, x_j)}$$

*Smaller average distortion implies greater similarity between $d_1$ and $d_2$.*

In many cases, the metric learned by the various algorithms will be a scalar multiple of the actual metric, so we will solve for the scale $\alpha := \arg\min_c \|D - c\hat{D}\|_F$, before calculating the average distortion.[7]

**Definition 25.** *Let $d$ be a metric on the nodes of a graph $G = (V, E)$. For $v \in V$, let $N(v) = \{u_1, \ldots, u_{deg(v)}\}$ be the neighborhood of $v$. Then let $B_{v,u_i} = \{u \in V \setminus \{u\} : d(u, v) \leq d(v, u_i)\}$. Then the mean average precision (MAP) is defined to be*

$$\frac{1}{n} \sum_{v \in V} \frac{1}{deg(v)} \sum_{i=1}^{|N(v)|} \frac{|N(v) \cap B_{v,u_i}|}{|B_{v,u_i}|}$$

*Closer MAP is to 1, the closer $d$ is to approximating $d_G$.*

### 12.2 TreeRep

There are a few practical details that must be discussed in relation to the TREEREP algorithm.

1. Pre-allocate the matrix for the weights of edges of the tree as a dense matrix. Doing this greatly speeds up computations. Note the proof of Lemma 2, show that we need at most $n$ Steiner nodes. Thus, the tree has about $2n$ nodes. Since the input to the algorithm is a dense $n \times n$ matrix, we already need $O(n^2)$ memory. Thus, having a dense $2n \times 2n$ matrix is still linear memory usage in the size of the input.
2. When doing zone 2 recursions pick the node closest to $r$ as the new $z$ as suggested by Proposition 1.
3. The placement of nodes into their respective zones can be done in parallel. For all of the experiments in the paper, we used 8 threads to do the placement for all of the experiments, except that we used 1 thread for the random points from $\mathbb{H}^k$ experiment and for CBMC experiment.
4. All of the numbers reported are averages over 20 iterations. We could have also picked the best over 20 iterations as our algorithm is fast enough for this to be viable.
5. When checking for equality, instead of checking for exact equality, we checked whether two numbers are within 0.1 of each other.
6. It is possible for some of the edge weights to be set to a negative number. In this case, after the algorithm terminated we set those edge weights to 0.

### 12.3 Bartal

We sample 200 trees from the distribution and compute the metric assuming that we are embedding into the distribution restricted to these 200 trees.

### 12.4 Neighbor Join

The following implementation of NJ was used: http://crsl4.github.io/PhyloNetworks.jl/latest/. We set the options so as to not have any negative edge weights.

## 12.5 MST

Prim's algorithm for calculating MST was used. We used the implementation at https://github.com/JuliaGraphs/LightGraphs.jl

## 12.6 LS

Low stretch spanning trees are calculated using Laplacian package in Julia. This code is based an adaptation of [2] by the authors of [16].

## 12.7 LevelTree and ConstructTree

To the best of the authors knowledge there does not exist a publicly available implementations of these algorithms. Both of these algorithms were implemented by the authors.

Note that LevelTree claims to be a $O(n)$ algorithm, but this only true, once we have calculated the sphere $S_n$ needed for the algorithm. However, it takes $O(n^2)$ time to calculate the spheres $S_n$ (equivalent to solving single source all destination shortest path problem).

## 12.8 PM and LM

The following options were used. The number of epochs was to set to be higher than default. Everything else was left at default. One note about PM and LM is that their objective function is set up to optimize for MAP and not average distortion.

1. -lr 0.3
2. -epochs 1000
3. -burnin 20
4. -negs 50
5. -fresh
6. -sparse
7. -train_threads 2
8. -ndproc 4
9. -batchsize 10

For PM we used `-manifold poincare`, for LM we used `-manifold lorentz`. The code is taken from https://github.com/facebookresearch/poincare-embeddings

## 12.9 PT

The following options were used. We used the `-learn-scale` option as based on the discussion in the appendix of Sala et al. [36] learning the scale results in better quality metrics. Additionally, we add a burnin phase to the optimization. Finally, based on the discussion in [36], the objective function for PT has a lot of shallow local minimas. Thus, we added momentum and used Adagrad for the optimization to try and avoid these local minimums.

1. --learn-scale
2. --burn-in 100
3. --momentum 0.9
4. --use-adagrad
5. --l 5.0
6. --epochs 1000
7. --batch-size 256
8. --subsample 64

The code is taken from https://github.com/HazyResearch/hyperbolics

## 12.10 Hardware

All experiments were run on Google cloud instances. For PM, LM and PT we created a fresh instance for each algorithm. Each instance for an algorithm only had the bare minimum installed to run those

algorithms. We used `n1-highmem-8` instances. The specification of each of the instances are as follows:

1. 8 cores each with 6.5 GB of ram.
2. Ubuntu-1604-xenial-v20190913 operting system.
3. 100 standard persistent disk.

For TreeRep, NJ, CT, LT and MST, we ran all code via a Jupyter notebook interface running Julia 1.1.0. All experiments (except for the experiments with Enron and Wordnet), we done on instances with the same specification as above.

For Enron and Wordnet, we need more memory to store the distance matrices. Thus, used an since with the following specifications.

1. 24 cores each with 6.5 GB of ram.
2. Ubuntu-1604-xenial-v20190913 operting system.
3. 100 standard persistent disk.

## 12.11 Synthetic $0$-hyperbolic metrics

To produce random synthetic 0-hyperbolic metrics, we do the following. First, we take a complete binary tree of depth $i$. We then compute its double tree. Then for each node in this tree we sample a number $C$ from 2 to 10 and replace the node with a clique of size $C$. We then pick a random node in the tree and compute the breadth first search tree from that node. We then assign edge uniformly randomly, sampled from $[0, 1]$.

## 12.12 Synthetic Data Sets

Here we sampled coordinates from the standard normal $\mathcal{N}(0, 1)$. The final coordinate $x_0$ is set so that the point lies on the hyperboloid manifold. In the presence of a scale we just multiplied each coordinate by that scale before calculating $x_0$. We ran TREEREP with 1 thread.

## 12.13 Phylogenetic and Single Cell Data

The immunological distances can be seen in Figure 5. The matrix is symmeterized by averaging across the diagonal. In this case, we ran TREEREP 10 times and picked the tree with the lowest average distortion.

The figures for the trees are produced using an adaptation of Sarkar's construction for Euclidean space. The code from PT also produces a picture. This picture can be seen in Figure 4. As we can see, this figure is similar to the one in the main text.

For the Zeisel data we did the same pre-processing as done in Dumitrascu et al. [14]. For PM and MST, we use 10 nearest neighbor graph. For LS we used the complete graph.

For the CBMC data we did the same pre-processing as done in Dumitrascu et al. [14]. For MST and LS we used the complete graph.

## 12.14 Unweighted Graphs

Some of the graphs are disconnected. The largest connected component of each graph was used.

For $\delta$ calculation, we normalized the distances so that the maximum distance was 1 and then calculated $\delta$. For Celegans, Diseasome, and Phds,, this calculation is exact.

For Yeast, Grid-worm and GRQC, we fixed the base point to be $w = 1$ and then calculated $\delta$. It is known from theory that for any fixed base point the $\delta$ is at least half of the $\delta$ for the whole metric [7]. Thus, we get the inequality.

All experiments with a "-" were terminated after 4 hours.

## 12.15 Calculating $\alpha$

Can be calculated directly using

Figure 4: Figure for Sarich data produced by PT code

```
dog          0  32  48  51  50  48  98 148
bear        32   0  26  34  29  33  84 136
raccoon     48  26   0  42  44  44  92 152
weasel      51  34  42   0  44  38  86 142
seal        50  29  44  44   0  24  89 142
sea lion    48  33  44  38  24   0  90 142
cat         98  84  92  86  89  90   0 148
monkey     148 136 152 142 142 142 148   0
```

Figure 5: Immunological distances from [37]

$$\alpha = \frac{Tr(D' * D)}{\|D\|_F^2}$$

## 13 Tree Representation Pseudo-code

We can see examples of what happens when we set the new Steiner node for the two different kinds of recursion in Figure 6

Table 7: Table with the time taken in seconds, MAP, and average distortion for all of the algorithms when given metrics that come from unweighted graph. Darker cell colors indicates better numbers for MAP and average distortion. The number next to PT, PM, LM is the dimension of the space used to learn the embedding. The numbers for TREEREP (TR) are the average numbers over 20 trials. The table also shows some graph statistics such as $n$, the number of nodes, $m$, the number of edges, and $\delta$, the hyperbolcity of the metric.

| Graph | | TR | NJ | MST | LT | CT | LS | Bartal | PT 2 | PT 200 | PM 2 | LM 2 | LM 200 | PM 200 |
|---|---|---|---|---|---|---|---|---|---|---|---|---|---|---|
| | $n$ | | | | | | MAP | | | | | | | |
| Celegan | 452 | 0.473 | 0.713 | 0.337 | 0.272 | 0.447 | 0.313 | 0.436 | 0.098 | 0.857 | 0.479 | 0.466 | 0.646 | 0.662 |
| Dieseasome | 516 | 0.895 | 0.962 | 0.789 | 0.725 | 0.815 | 0.785 | 0.610 | 0.392 | 0.868 | 0.799 | 0.781 | 0.874 | 0.886 |
| CS Phd | 1025 | 0.979 | 0.993 | 0.991 | 0.964 | 0.807 | 0.991 | 0.190 | 0.190 | 0.556 | 0.537 | 0.537 | 0.593 | 0.593 |
| Yeast | 1458 | 0.815 | 0.892 | 0.871 | 0.742 | 0.859 | 0.873 | - | 0.235 | 0.658 | 0.522 | 0.513 | 0.641 | 0.643 |
| Grid-worm | 3337 | 0.707 | 0.800 | 0.768 | 0.657 | - | 0.766 | - | - | - | 0.334 | 0.306 | 0.558 | 0.553 |
| GRQC | 4158 | 0.685 | 0.862 | 0.686 | 0.480 | - | 0.684 | - | - | - | 0.589 | 0.603 | 0.783 | 0.784 |
| Enron | 33695 | 0.570 | - | 0.524 | - | - | 0.523 | - | - | - | - | - | - | - |
| Wordnet | 74374 | 0.984 | - | 0.989 | - | - | 0.989 | - | - | - | - | - | - | - |
| | $m$ | | | | | | Average Distortion | | | | | | | |
| Celegan | 2024 | 0.197 | 0.124 | 0.255 | 0.166 | 0.325 | 0.353 | 0.220 | 0.236 | 0.096 | 0.236 | 0.249 | 0.224 | 0.211 |
| Dieseasome | 1188 | 0.188 | 0.161 | 0.161 | 0.157 | 0.315 | 0.228 | 0.330 | 0.227 | 0.05 | 0.323 | 0.328 | 0.335 | 0.332 |
| CS Phd | 1043 | 0.204 | 0.134 | 0.298 | 0.161 | 0.282 | 0.291 | 0.326 | 0.295 | 0.105 | 0.374 | 0.378 | 0.378 | 0.380 |
| Yeast | 1948 | 0.205 | 0.149 | 0.243 | 0.243 | 0.282 | 0.243 | - | 0.230 | 0.089 | 0.246 | 0.248 | 0.234 | 0.234 |
| Grid-worm | 6421 | 0.188 | 0.135 | 0.171 | 0.202 | - | 0.234 | - | - | - | 0.196 | 0.203 | 0.192 | 0.193 |
| GRQC | 13422 | 0.192 | 0.200 | 0.275 | 0.267 | - | 0.206 | - | - | - | 0.212 | 0.198 | 0.193 | 0.193 |
| Enron | 180810 | 0.453 | - | 0.607 | - | - | 0.562 | - | - | - | - | - | - | - |
| Wordnet | 75834 | 0.131 | - | 0.336 | - | - | 0.071 | - | - | - | - | - | - | - |
| | $\delta$ | | | | | | Time in seconds | | | | | | | |
| Celegan | 0.21 | 0.014 | 0.28 | 0.0002 | 0.086 | 0.9 | 0.001 | 226 | 573 | 1156 | 712 | 523 | 1578 | 1927 |
| Dieseasome | 0.17 | 0.017 | 0.41 | 0.0003 | 0.39 | 15.76 | 0.001 | 313 | 678 | 1479 | 414 | 365 | 978 | 1112 |
| CS Phd | 0.23 | 0.037 | 2.94 | 0.0007 | 1.97 | 226 | 0.006 | 3559 | 1607 | 4145 | 467 | 324 | 768 | 1149 |
| Yeast | $\leq 0.32$ | 0.057 | 8.04 | 0.0008 | 8.21 | 957 | 0.001 | - | 9526 | 17876 | 972 | 619 | 1334 | 2269 |
| Grid-worm | $\leq 0.38$ | 0.731 | 163 | 0.001 | 191 | - | 0.007 | - | - | - | 2645 | 1973 | 4674 | 5593 |
| GRQC | $\leq 0.36$ | 0.42 | 311 | 0.0014 | 70.9 | - | 0.006 | - | - | - | 7524 | 7217 | 9767 | 1187 |
| Enron | - | | 27 | - | 0.013 | - | - | 0.13 | - | - | - | - | - | - | - |
| Wordnet | - | | 74 | - | 0.18 | - | - | 0.08 | - | - | - | - | - | - | - |

---

**Algorithm 3** Recursive parts of TreeRep.

---

1: **function** ZONE1_RECURSION($T, d_T, d, L, v$)
2:     **if** Length($L$) == 0 **then**
3:         **return** $T$
4:     **if** Length($L$) == 1 **then**
5:         Set $u$ = pop($L$) and add edge $(u, v)$ to $E$
6:         Set edge weight $d_T(u, v) = d(u, v)$
7:         **return** $T$
8:     Set $u =$ pop($L$), $z =$ pop($L$)
9:     **return** RECURSIVE_STEP($T, L, v, u, z, d, d_T$)
10:
11: **function** ZONE2_RECURSION($T, d_T, d, L, u, v$)
12:     **if** Length($L$) == 0 **then return** $T$
13:     Set $z=$ the closest node to $v$.
14:     Delete edge $(u, v)$
15:     **return:** RECURSIVE_STEP($T, L, v, u, z, d, d_T$)

---

**Algorithm 4** Metric to tree structure algorithm.

---

1: **function** TREE STRUCTURE(X, $d$)
2:     $T = (V, E, d') = \emptyset$
3:     Pick any three data points uniformly at random $x, y, z \in X$.
4:     $T$ = RECURSIVE_STEP($T, X, x, y, z, d, d_T$,)
5:     **return** $T$

6:

7:

8: **function** RECURSIVE_STEP($T, X, x, y, z, d, d_T$,)
9:     Let $Z1(r \to [], x \to [], y \to [], z \to []), Z2(x \to [], y \to [], z \to [])$    // Dictionaries of list for various zones
10:     Place an additional node $r$ in $V$ and add edges $xr, yr, zr$ to $E$
11:     Set the weights $d_T(x, r) = (y, z)_x$, $d_T(y, r) = (x, z)_y$, and $d_T(z, r) = (x, y)_z$ // If edge weight = 0, contract the edge.
12:     **for** all remaining data points $w \in X$ **do**
13:         $a = (x, y)_w$, $b = (y, z)_w$, $c = (z, x)_w$, $m = 0$, $m2 = 0$
14:         **if** $a == b == c$ **then**
15:             push($w, Z1[r]$)
16:             Set $d_T(w, r) = (x, y)_w$
17:         **else if** $a == maximum(a, b, c)$ **then**
18:             $\pi = (x \to z, y \to y, z \to x)$
19:             $m = b$, $m2 = c$
20:             Set $d_T(w, r) = a$
21:         **else if** $b == maximum(a, b, c)$ **then**
22:             $\pi = (x \to x, y \to y, z \to z)$
23:             $m = a$, $m2 = c$
24:             Set $d_T(w, r) = b$
25:         **else if** $c == maximum(a, b, c)$ **then**
26:             $\pi = (x \to y, y \to x, z \to z)$
27:             $m = a$, $m2 = b$
28:             Set $d_T(w, r) = c$
29:         **if** $d(w, \pi x) == m$ or $d(w, \pi x) == m2$ **then**
30:             push($w, Z1[\pi x]$)
31:         **else**
32:             push($w, Z2[\pi x]$)
          // recurse on each of the zones
33:     $T$ = ZONE1_RECURSION($T, d_T, d, Z1[r], r$)
34:     $T$ = ZONE1_RECURSION($T, d_T, d, Z1[x], x$)
35:     $T$ = ZONE1_RECURSION($T, d_T, d, Z1[y], y$)
36:     $T$ = ZONE1_RECURSION($T, d_T, d, Z1[y], z$)
37:     $T$ = ZONE2_RECURSION($T, d_T, d, Z2[x], x, r$)
38:     $T$ = ZONE2_RECURSION($T, d_T, d, Z2[y], y, r$)
39:     $T$ = ZONE2_RECURSION($T, d_T, d, Z2[z], z, r$)
        **return** $T$

---

Figure 6: Figure showing the placement of the Steiner node $R'$ for the Zone 1 and Zone 2 recursion. The nodes in orange are Steiner nodes and the nodes in green come from the data set $V$.

## Footnotes

[7]For NJ and LT, computing this $\alpha$ made the average distortion worse, so we report numbers un-scaled. Additionally, computing $\alpha$ is too computationally expensive for bigger data sets and was not done for the Enron and Wordnet data set.