[Reviews · NeurIPS 2020]

Review 1

Summary and Contributions: The paper proposes TreeRep, an algorithm that constructs a tree-metric approximation of a Gromov hyperbolic metric. This algorithm runs thousands of times faster than existing approaches for embedding into hyperbolic manifolds.

Strengths: The algorithm is interesting and a (I think) novel approach for metric learning in this space. The paper is cleanly written and well motivated. The algorithm is relatively easy to understand and comes with some nice theoretical guarantees.

Weaknesses: The relationship of the method to hyperbolic space is not obvious. It seems to me that recovering a tree from a metric isn't inherently a hyperbolic-space thing. Sure, if you then proceeded to embed the tree into hyperbolic space using Sarkar's construction, that would make the relationship clear...but as far as I can tell this wasn't done. The paper would be a stronger contender in this category if the method were used to produce hyperbolic embeddings that were then used for some downstream task (such as MAP) rather than just evaluating distortion.

Correctness: As far as I can tell, the claims are correct and the method is valid. The empirical methodology looks to be the correct one, although it is limited to only looking at distortion (rather than also evaluating other downstream task metrics for an embedding).

Clarity: The paper is fairly clear, although I think it would be better to cut down on the definitions on page 3 (right now it reads as a list-of-definitions section). E.g. definition 4 could be cut as it appears to not really be used. Or, definition 7 is well-known and we can just assume people know what a geodesic is. You could then spend the space on having an actual algorithm block for TreeRep, which would make it easier to follow what you are doing.

Relation to Prior Work: The related work section seems complete and detailed. I have some concerns about the comparison to Sala et al [36]. Sala et al [36] presents an algorithm, h-MDS (their Algorithm 2) which gives guaranteed-exact recovery of a metric embeddable exactly in hyperbolic space. The fact that [36] gives such an algorithm with guarantees makes the description in this paper "They do not, however, come with rigorous geometric guarantees of the optimal solutions" seem kinda misleading as applied to [36]. More worryingly, although the empirical experiments in this paper seem to compare to [36], they do not appear to try h-MDS at all: even though h-MDS should get zero distortion on its random-points-on-a-hyperbolic-manifold test. It is not clear why this comparison was not done.

Reproducibility: Yes

Additional Feedback: Your graph legend formatting in Figure 2 is messed up.


Review 2

Summary and Contributions: In this paper, the authors preset a new method for learning hyperbolic representations. The authors present an algorithm, TreeRep, to learn a tree structure on the data as an intermediate step before determining the hyperbolic embedding.

Strengths: The paper and and its appendix include theoretical guarantees that their algorithm does return a tree and several structural lemmas. Empirically, their algorithm also has lower average distortion and higher mean average precision than previous algorithms.

Weaknesses: My only concern is that I'm not sure if it will be something the larger NeurIPS community might be interested in.

Correctness: I read the relevant proofs of Theorem 1 and it seemed correct to me.

Clarity: The paper is well-written. The definitions and problems are clearly stated as well as the theorem and the proof. My only comment is that I did not like the sentence "Furthermore the algorithm is embarrassingly parallelizable" as part of the theorem. It probably should be a remark when the implementation is discussed.

Relation to Prior Work: Yes.

Reproducibility: Yes

Additional Feedback:


Review 3

Summary and Contributions: The authors propose an efficient algorithm to compute low-dimensional hyperbolic embedding of the data via a tree structure.

Strengths: The authors provide extensive experiments to evaluate their method.

Weaknesses: The logic of writing and typos make the paper somehow confusing. A clearer problem statement and motivations are expected.

Correctness: The problem itself is worth studying and most of the statements are correct.

Clarity: I have to say some parts of the paper is really confusing to follow. I like the topic and the solid algorithm work, but it is better if the authors can refine this paper and give a clearer explanation and comparisons.

Relation to Prior Work: Many related work is mentioned. But I cannot say they are discussed in a clear way.

Reproducibility: Yes

Additional Feedback: (1) In Abstract, you mentioned "rather than determining the low-dimensional hyperbolic embedding directly, we learn a tree structure...", but the title of your paper is "Tree! I am no Tree! I am a low dimensional hyperbolic embedding". This is confusing, please clarify what do you want to express. (2) In Introduction, you mentioned your method is "extremely fast algorithm" in learning a tree structure to approximate the metric. I see Table 1, 2 and 3 contain the results of testing computational efficiency. But it seems the efficiency is not that impressive. Please give explanations on that statement in Introduction. (3) I notice you bring up Yu and De Sa's work in NeurlPS2019, I wonder if a comparison of their method and yours can be made, since you are towards solving a similar issue. (4) what are T hat and dThat mean in Lemma 2? "Then we can extend..." This sentence on line 152 is confusing. (5) Definition 10 is confusing. "Let us defining the following two zone types", but three items are listed. please clearly define zone1 and zone2 ######################## Thanks the feedback from the author. I like what the authors want to deliver in this paper, but I would not change the score. Logically, this submission required to be further clarified about the problem statement, motivation and contributions. I was confused by several expressions in the paper (eg. my comment 4 and 5), especially the introduction part, but still I think it is unclear to well follow the paper after reading the feedback. I understand the limitation of feedback pages would force the authors to give up answering some questions, but I also believe more work is supposed to be done before submitting it officially. Besides the logic and expression of the paper, I don't think the authors have well discusses about the prior work. Discussions about Yu and De Sa’s work in feedback does not make sense to me. Since the authors have pointed out the location of the code, I changed the reproducibility to Yes.


Review 4

Summary and Contributions: This paper suggests an algorithm TreeRep that, take a delta-hyperbolic metric space as an input and learns a tree structure with low distortion. For the case when delta = 0, TreeRep has a theoretical guarantee that the output tree recovers the input metric. Also, it is empirically shown that TreeRep is faster and has lower distortion than previous algorithms.

Strengths: As far as I checked, the theoretical claims in this paper are solid, and also experimentally verified. I think practitioners will find TreeRep useful as a method to embed data into trees.

Weaknesses: TreeRep provides a guarantee that when tree is given as an input then the output recovers the tree metric, but for general delta-hyperbolic metric space as an input, Proposition 1 provides a bound of the distortion given 3 points but not giving the global bound on the distortion. I think it should be clearly discussed or mentioned that how the result in Proposition 1 is different from the global bound on the distortion and how it can be possibly extended to give a bound for the global bound on the distortion. Also, in the experiments, NJ outperforms TreeRep in the hyperbolic manifold dataset and TreeRep outperforms NJ in the biological data in terms of the distortion. So I think it would be helpful if the authors can give some intuition on under what situations TreeRep outperforms other tree construction methods.

Correctness: In Definition 5 (p.3, line 97-99), I guess $d(( e, t_1 ), ( e, t_2 )) = w(e) | t_1 - t_2 |$, since in the current definition $d(( e, t ), ( e, t ))$ becomes infinite. Also, when constructing the metric graph X, you should quotient as X = E\times[0,1]/\sim$, where (e_1, 0) ~ (e_2, 0) if e_1 = (v, v_1) and e_2 = (v, v_2) (and similar for (e_1, 0) ~ (e_2, 1), (e_1, 1) ~ (e_2, 0), (e_1, 1) ~ (e_2, 1) as well). And you should also define how the distance should be defined between ( e_1, t_1 ) and ( e_2, t_2 ). Also in Supplement, p.5, line 534: "Let $x, y \in X$ and let $r \in g(x, y)$" -> "Let $x, y \in X$, then $r \in g(x, y)$"

Clarity: I think the paper is overall clearly written, except that some definitions and statements are not fully clearly written as I mentioned in Correctness.

Relation to Prior Work: I think the comparison to related work is well discussed in Section 1.

Reproducibility: Yes

Additional Feedback: Minor typos: p.2, line 59: "take as an input graphs, not metrics" -> "take as input graphs, not metrics" p.3, line 95: "a a weighted tree" -> "a weighted tree" p.3, line 116: "a hyperbolic representations" -> "hyperbolic representations" p.4, line 130: "that that" -> "that" p.4, line 155: "let us defining" -> "let us define" p.7, line 254: "the we" -> "we" Supplement, p.4, line 525: "let us defining" -> "let us define" Supplement, p.17, line 617: "$\pi y, \pi z$" -> "$\pi y, \pi z$." I read the authors' feedback and I agree with their comments, so I increase the overall score to 6. As I have suggested, I think it would be helpful to highlight how the suggested method differs from and is better than existing approaches.

[Author Response · NeurIPS 2020]



| Data | MAP | | | Distortion | | | Time | | |
|---|---|---|---|---|---|---|---|---|---|
| | TR | HMDS 2 | HMDS 200 | TR | HMDS 2 | HMDS 200 | TR | HMDS 2 | HMDS 200 |
| Celegans | 0.473 | 0.110 | 0.400 | 0.197 | 0.527 | 0.142 | 0.014 | 0.03 | 39 |
| Diseaseome | 0.895 | 0.284 | 0.833 | 0.188 | 0.427 | 0.160 | 0.017 | 0.03 | 56 |
| PHD | 0.979 | 0.027 | 0.203 | 0.204 | 0.461 | 0.233 | 0.037 | 0.06 | 11 |
| Yeast | 0.815 | 0.147 | 0.386 | 0.205 | 0.166 | 0.162 | 0.057 | 0.05 | 74 |
| Grid worm | 0.707 | 0.006 | 0.285 | 0.188 | 0.494 | 0.132 | 0.731 | 0.5 | 381 |
| GRQC | 0.685 | 0.142 | 0.482 | 0.192 | 0.460 | 0.137 | 0.42 | 0.86 | 512 |

Thank you for carefully reviewing our submission.

R3"In Abstract, ... please clarify what do you want to express." **Ans: See next discussion.**

R1"The relationship of the method to hyperbolic space ... Sarkar's construction, that would make the relationship clear"
**Ans: Trees and hyperbolic manifolds are inherently linked via their geometry.** We discuss the history of finding a
discrete, combinatorial structure to represent the metric first and then embedding it into hyperbolic space. Sections 8, 9
in the Appendix, describe this link explicitly and mathematically. For example, in section 8, we present a well known
fact from hyperbolic geometry that the asymptotic cone (a geometric tool to study large scale geometry) of a hyperbolic
manifold is a tree. Furthermore, references such as "On tree-likeness of hyperbolic space"[20] present different types of
geometric connections. Finally, we note that Sarkar's method gives us a way to embed a tree into hyperbolic space with
as low distortion as we desire (at the cost of applying a rescaling of the original metric, followed by unscaling the
embedded metric - this scale trick is used extensively by De Sa [36]).

R1"h-MDS (their Algorithm 2) ... hyperbolic space." **Ans:** For comparison purposes, we chose the other algorithm
from [36] and not hmds because hmds has guaranteed good performance only if **the metric comes from a hyperbolic**
**manifold** and we use hmds to embed into **same dimension or higher**. For completeness and to address the reviewers
comment, we have now compared against hmds and the results can be seen in the figures and the table above. The only
result not displayed is the performance on the bio data sets as hmds either resulted in infs or returned all 0 metric. In
the legend above, HMDS 2 uses hmds to embed into 2 dimensional manifold, HMDS 200 is an embedding into a 200
dimensional manifold, and hdms is embedding into the same dimension as the original points. As we can see, for all
experiments **HMDS 2 does not perform well** and the scaling of the distances has an impact numerically. HMDS 200
also has bad MAP, but good distortion. With regards to computational efficiency, HMDS 2 was faster than all except
TreeRep and MST, while HMDS 200 was slower than all tree methods but faster than the optimization based methods.
These results will be added to the final version. The code used to run hmds is from authors of [36]'s github repository.

R4"TreeRep provides a guarantee ... global bound on the distortion." **Ans:** We are missing an overall global bound on
the distortion. We leave it as future (technical theoretical) work. We verify experimentally that we have low global
distortion.

R4"Also, in the experiments...TreeRep outperforms other tree construction methods." **Ans:** The addition of Steiner
nodes gives us more flexibility in creating trees over methods such as MST and low-stretch trees. We conjecture that the
reason we outperform CT [1] is this algorithm uses only 3 points when placing of nodes, but the Gromov condition
depends on 4 points and so the algorithm misses some geometric information. NJ is a local to global method; i.e., it
makes local connections between points, whereas we are global to local. We do not know a priori, however, which
method, NJ or TreeRep, will work better. Nevertheless, **our method is faster than NJ and can always be used.**

R2"My only concern ... NeurIPS community might be interested in." **Ans:** A variety of the papers on this topic have
been published in Neurips/ICML with great success [30,31,36,43]

R3"Reproducibility...No" **Ans:** We provide code, pseudo-code, set ups, parameters, our data, and steps for how to run
our experiments at the link in Footnote 4 on page 6 of the submission.

R1"the paper would be a stronger contender...downstream task (such as MAP)" **Ans:** For unweighted graphs, results
in table 6 of the Appendix, we do compute MAP. The last two paragraphs of the paper discuss this experiment.

R4"In Definition 5 ..." **Ans:** Yes, that was a mistake, we have fixed the definition.

R3"Table 1, 2 and 3...efficiency is not that impressive" **Ans:** The tables do show our efficiency. Table 1 shows that for
$n = 1611$, we are 2 orders of magnitude faster than NJ. Table 3 shows that we are at least an one order magnitude faster
than all algorithms except MST (which performs poorly) and many orders of magnitude faster than PT and PM.

R3"Yu and De Sa's work in NeurIPS2019... " **Ans:**. Yu and De Sa primarily proposes a method of representing points
on the hyperbolic manifold using a small number of bits rather than a new method to learn a representation. Indeed,
they use the learning methods from [30,31] and use their proposed representation to avoid numerical issues. Thus, we
do not directly compare against them

[Meta-Review · NeurIPS 2020]

This paper presents a novel method for embedding a set of data points into a tree structure, maximally preserving a given metric. The reviewers think the methodology is sound, and experimental results demonstrate its efficiency. The authors are encouraged to improve the presentation and organization as suggested by multiple reviewers, to increase its readability and bring out the key intuitions.